# Symbiotic polyamine metabolism regulates epithelial proliferation and macrophage differentiation in the colon

Atsuo Nakamura [1,2], Shin Kurihara[3,10], Daisuke Takahashi [1], Wakana Ohashi[1], Yutaka Nakamura[1], Shunsuke Kimura[1,4], Masayoshi Onuki[1], Aiko Kume[2], Yukiko Sasazawa[5], Yukihiro Furusawa[1,11], Yuuki Obata [1,12], Shinji Fukuda [4,6,7,8], Shinji Saiki[5], Mitsuharu Matsumoto [2✉] & Koji Hase [1,9✉]

Intestinal microbiota-derived metabolites have biological importance for the host. Polyamines, such as putrescine and spermidine, are produced by the intestinal microbiota and regulate multiple biological processes. Increased colonic luminal polyamines promote longevity in mice. However, no direct evidence has shown that microbial polyamines are incorporated into host cells to regulate cellular responses. Here, we show that microbial polyamines reinforce colonic epithelial proliferation and regulate macrophage differentiation. Colonisation by wild-type, but not polyamine biosynthesis-deficient, *Escherichia coli* in germ-free mice raises intracellular polyamine levels in colonocytes, accelerating epithelial renewal. Commensal bacterium-derived putrescine increases the abundance of anti-inflammatory macrophages in the colon. The bacterial polyamines ameliorate symptoms of dextran sulfate sodium-induced colitis in mice. These effects mainly result from enhanced hypusination of eukaryotic initiation translation factor. We conclude that bacterial putrescine functions as a substrate for symbiotic metabolism and is further absorbed and metabolised by the host, thus helping maintain mucosal homoeostasis in the intestine.

[1] Division of Biochemistry, Faculty of Pharmacy and Graduate School of Pharmaceutical Science, Keio University, Minato-ku, Tokyo, Japan. [2] Dairy Science and Technology Institute, Kyodo Milk Industry Co., Ltd., Hinode-machi, Nishitama-gun, Tokyo, Japan. [3] Faculty of Bioresources and Environmental Sciences, Ishikawa Prefectural University, Nonoichi, Ishikawa, Japan. [4] PRESTO, Japan Science and Technology Agency, Kawaguchi, Saitama, Japan. [5] Department of Neurology, Graduate School of Medicine, Juntendo University, Bunkyo-ku, Tokyo, Japan. [6] Institute for Advanced Biosciences, Keio University, Tsuruoka, Yamagata, Japan. [7] Transborder Medical Research Center, University of Tsukuba, Tsukuba, Ibaraki, Japan. [8] Intestinal Microbiota Project, Kanagawa Institute of Industrial Science and Technology, Kawasaki, Kanagawa, Japan. [9] International Research and Development Center for Mucosal Vaccines, The Institute of Medical Science, The University of Tokyo (IMSUT), Bunkyo-ku, Tokyo, Japan. [10]Present address: Faculty of Biology-Oriented Science and Technology, Kindai University, Kinokawa, Wakayama, Japan. [11]Present address: Department of Liberal Arts and Sciences, Toyama Prefectural University, Kurokawa, Toyama, Japan. [12]Present address: The Francis Crick Institute, London, UK. ✉email: m-matumoto@meito.co.jp; hase-kj@pha.keio.ac.jp

Polyamines, which are aliphatic compounds containing more than two amino groups, are widely distributed among prokaryotes and eukaryotes, including humans, and regulate multiple biological processes, including translation, transcription and cell proliferation and differentiation[1,2]. Among polyamines, putrescine and spermidine are especially prevalent in the large intestine of human and mice[3,4]. We previously reported that the abundance of luminal polyamines is positively correlated with longevity in mice[5]. Likewise, exogenous spermidine administration extends the lifespan in model organisms and improves cardiac dysfunction as well as metabolic syndrome in mice by inducing autophagy[6–8]. Furthermore, the enhancement of luminal putrescine production by administration of probiotics and arginine improved reactive hyperaemia index representing endothelial function in randomised controlled clinical trial[9]. These observations illustrate that exogenous polyamines may possess a substantial impact on the host physiology.

In mammalian cells, putrescine is biosynthesized by ornithine decarboxylase (ODC) from precursor ornithine, then spermidine is generated from putrescine by addition of aminopropyl groups derived from decarboxylated S-adenosyl methionine. Spermidine serves as a substrate for hypusine, in which the aminobutyl group of spermidine is attached to a specific lysine residue. Hypusination of eukaryotic translation initiation factor 5 A (eIF5A) is required for its activation[10–13], and hypusinated eIF5A (hyp-eIF5A) is involved in translation initiation, elongation, termination, and cell cycles[10–12].

The intestinal epithelium is one of the most rapidly self-renewing tissues in adult mammals, with cell turnover every 3–5 days[14]. Microbial colonisation induces rapid epithelial turnover in the intestine[15,16]. Because intestinal epithelial cells are constantly exposed to pathogens and harmful components from diets or microbiota, this rapid turnover promotes elimination of infected and/or damaged cells[17–19]. Previous studies demonstrated the importance of polyamine for proliferation and wound healing in the intestinal epithelium[20,21], whereas contribution of microbial polyamines to epithelial cell turnovder has not been to directly explored.

Polyamines also regulate immune responses. For example, spermidine restores CD8$^+$ T cell responses in elderly mice[22]. Further, the treatment with spermine inhibits lipopolysaccharide (LPS)-mediated production of nitric oxide and pro-inflammatory cytokines, such as tumour necrosis factor (TNF), interleukin (IL)-1β, and IL-6 in mouse macrophages and human mononuclear cells[23,24]. Recent studies have demonstrated that deleting ODC enhances classical activation of M1 macrophages[25]. Conversely, polyamine synthesis is enhanced in alternatively activated M2 macrophages by IL-4 stimulation[26]. Compared with other organs, larger numbers of macrophages reside in the intestines[27], accumulating in the colonic lamina propria (cLP) near the epithelial layer[28]. Under physiological conditions, there are two major macrophage/monocyte subsets, inflammatory CX$_3$CR1$^{low}$Ly6C$^+$ cells and anti-inflammatory CX$_3$CR1$^{high}$Ly6C$^-$ cells, in the cLP[29]. While inflammatory CX$_3$CR1$^{low}$Ly6C$^+$ cells are involved in biological defences in the intestinal mucosal barrier, anti-inflammatory CX$_3$CR1$^{high}$Ly6C$^-$ cells play roles in immune tolerance to suppress excessive immune responses. However, it remains unknown whether polyamines influence the development of these two myeloid populations in the colon.

In this study, we evaluated the biological significance of commensal bacterium-derived polyamines. We established a gnotobiotic mice associated with the putrescine synthesis genes-sufficient or -deficient *Escherichia coli*[30]. We provide evidence that polyamines derived from colonic lumen were absorbed and utilised by the host, leading to enhanced the proliferation of colonic epithelial cells (CECs) and alleviated the development of inflammatory macrophages in the cLP. Thus, bacterial polyamines most likely contribute to the maintenance of intestinal homoeostasis.

## Results and discussion

**Microbial metabolites are associated with epithelial cell turnover in the colon.** Faecal microbial transplantation from specific-pathogen-free (SPF) mice to germ-free (GF) mice caused a hyperproliferative response in CECs, which peaked on day 3 (Supplementary Fig. 1a, b). CECs positive for the proliferative marker, Ki67, were decreased in the exGF mice from day 6 onward but remained twice that of the GF mice. We used metabolomic analysis of the caecal contents (luminal metabolites) and CECs (intracellular metabolites) to study the microbial products contributing to the hyperproliferative response (Supplementary Fig. 1c, d and Supplementary Data 1, 2). Colonocyte proliferation was positively correlated with 45 luminal metabolites and 15 intracellular metabolites (Supplementary Fig. 1e), among which, 4 metabolites (i.e., β-alanine, 5-aminovalerate, glutarate, and putrescine) were commonly detected in both analyses (Supplementary Fig. 1c–e). Increased intracellular polyamine levels, including those of putrescine, have been associated with cell proliferation in rapidly growing cells and tissues as well as in cancer[31–33]. Additionally, putrescine levels and biosynthesis are greatly upregulated in the ileal mucosa after a jejunectomy and in the duodenal mucosa after stress-induced damage in rats[34,35]. We observed prominently lower luminal polyamine levels in the GF mice than in the SPF and exGF mice, peaking on day 3 (Fig. 1a and Supplementary Fig. 1f, g). Because dietary polyamines are absorbed in the upper gastrointestinal tract and spread throughout the body[36,37], the major source of the lower luminal polyamines was likely derived from intestinal microbiota.

**Luminal bacterium-derived polyamines facilitate colonic epithelial proliferation.** Based on our observations and results from previous studies, we hypothesised that commensal bacterium-derived polyamines may be incorporated into colonocytes and may partly contribute to hyperproliferative responses. We evaluated this hypothesis using a gnotobiotic mouse system with putrescine synthesis-deficient or -sufficient bacteria. Certain commensal bacteria, such as *E. coli*, produce putrescine via two pathways using arginine and ornithine as substrates (Fig. 1b)[38]. Although *E. coli* is a minor member of the intestinal microbiota in adult humans, this species is prominently enriched in the microbial community early in life when the intestine is rapidly maturing[39,40]. To assess the contribution of bacterial putrescine to epithelial proliferation, we used *E. coli* SK930[30], which lacks genes for putrescine synthesis (Fig. 1b)[41,42]. Putrescine was nearly absent in the SK930 culture supernatant, and spermidine concentrations in the wild-type (WT) and SK930 culture supernatants were comparable to that in the Luria-Bertani (LB) medium (Supplementary Fig. 2a) with no defects in growth or LPS synthesis (Supplementary Fig. 2b, c). We inoculated the WT and SK930 strains into GF mice (F0), then generated F1 gnotobiotic mice via maternal-infant transmission to determine the effect of bacterial putrescine on the intestinal physiology (Fig. 1c). We obtained gnotobiotic mice with and without bacterium-derived putrescine (Fig. 1d), in which the number of *E. coli* in the luminal contents were comparable throughout the jejunal lumen to the distal colonic lumen in both mouse groups, as were the levels of other metabolites, including short-chain fatty acids (SCFAs; Supplementary Fig. 3a–c and Supplementary Table 1). Notably, WT-strain-associated, but not SK930-strain-associated, mice exhibited significantly increased intracellular levels of putrescine and its derivative, spermidine, in CECs (Fig. 1e),

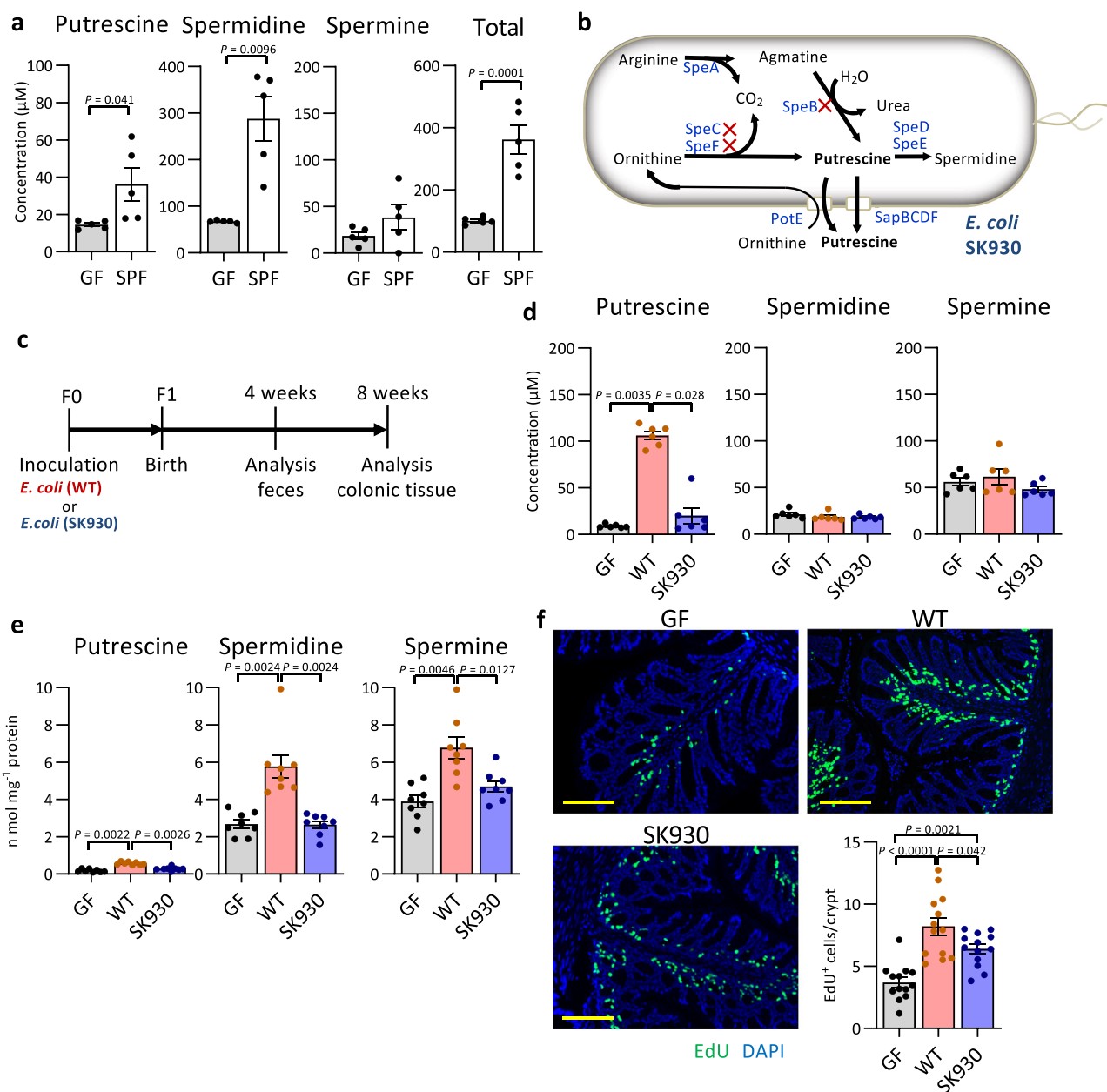

**Fig. 1 Bacterial putrescine facilitates epithelial cell proliferation in the colon. a** Polyamine concentrations in faeces of eight-weeks old GF and SPF mice ($n = 5$). **b** Polyamine synthesis pathway in *E. coli* and the deleted genes (marked with X) in an *E. coli* strain SK930. Blue letters represent enzymes and transporters: arginine decarboxylase (SpeA), agmatine ureohydrolase (SpeB), ornithine decarboxylase (SpeC and SpeF), adenosylmethionine decarboxylase (SpeD), spermidine synthase (SpeE), putrescine-ornithine antiport transporter (PotE) and putrescine exporters (SapBCDF). **c** Experimental design for gnotobiotic mice colonised with WT or SK930 strain. **d** The polyamine concentrations in faeces of F1 mice. **e** The polyamine concentrations in CECs ($n = 8$, independent animals). **f** Representative microscopic images of EdU assay and quantification of EdU-positive cells/crypt in each group. EdU-positive cells (green). Nuclei are counterstained with DAPI (blue) ($n = 8$, independent animals). Scale bars: 100 μm. Three randomly selected areas were examined per slide ($n = 13$). All data shown represent the mean ± SEM. Statistical significance was calculated using **a** the Welch's *t*-test (two-sided), **d**, **e** Kruskal–Wallis followed by Steel–Dwass post hoc test, **f** one-way ANOVA followed by Tukey's post-hoc test.

implying that bacterial putrescine is metabolised to spermidine in CECs. Moreover, CEC proliferation was significantly enhanced in WT-strain-associated mice compared with that in the GF and SK930-strain-associated mice (Fig. 1f and Supplementary Fig. 3d). SK930 colonisation also slightly increased the CEC proliferation likely due to the stimulation from LPS and/or lactate (Supplementary Table 1), both of which have been reported to accelerate CEC proliferation[43,44].

**CECs actively utilise luminal putrescine**. To confirm whether dietary putrescine reaches the colon, mice were orally administered stable-isotope-labelled $^{15}N_2$-putrescine in phosphate-buffered saline (PBS) containing blue food colouring agent as a marker to determine the position of the labelled-putrescine in the digestive tract. We analysed the isotope ratio of polyamines in the blue-stained luminal contents on arrival in the ileum (2 h after oral administration) and in the colon (3.5 h after oral

administration; Supplementary Fig. 4a, b). The labelled-putrescine/total-putrescine ratio was 35.0% in the ileal luminal contents 2 h after oral administration. Conversely, only minimal labelled putrescine (0.36%) was detected in the colonic luminal contents after 3.5 h (Supplementary Fig. 4c). These results demonstrated that a large portion of dietary putrescine was absorbed in the upper digestive tract before reaching the colonic lumen. To assess the uptake of luminal putrescine and its conversion to spermidine in CECs, we injected labelled putrescine into the colonic lumens of mice (luminal administration model; Supplementary Fig. 4d). Two hours post-injection, we compared the isotope ratios of the polyamines in the CECs with those in the CECs collected 2 and 3.5 h after oral administration, when the putrescine solution reached the ileum and colon, respectively (oral-administration model; Supplementary Fig. 4a, c). Isotope analysis revealed that the labelled-putrescine/total-putrescine ratio was 11.2% in the CECs in the luminal-administration model (Supplementary Fig. 4e). Conversely, the labelled-putrescine/total-putrescine ratios were only 2.96% and 2.23% in the CECs 2 and 3.5 h after oral administration, respectively. Additionally, labelled spermidine was higher in the luminal-administration-model CECs than in the oral-administration-model CECs (Supplementary Fig. 4e). However, 2 h after oral administration, when the putrescine had reached the ileum but not the colon, low levels of labelled putrescine were detected in the CECs, indicating that the oral putrescine may have reached the CECs through the bloodstream. Only a portion of the dietary putrescine was transported to the CECs, suggesting that exogenous local putrescine in the colonic lumen is actively incorporated and used as a polyamine-metabolic source for CECs.

Labelled putrescine was detected in the portal vein of the luminal-administration model (Supplementary Fig. 4f). Thus, intestinal bacterium-derived putrescine may affect the whole body. To further confirm that CECs use exogenous polyamines, we incubated intestinal organoids generated from colonic crypts in a standard medium[45] supplemented with 100 μM $^{15}N_2$-putrescine for 2 days. The intracellular isotope-labelled polyamines, $^{15}N_2$-putrescine and $^{15}N_2$-spermidine, occupied 47 and 30%, respectively, of the corresponding polyamines in the organoids (Fig. 2a, b). Thus, exogenous putrescine was actively uptaken and converted to spermidine in the CECs. Importantly, we detected putrescine at $132.3 \pm 17.3$ μM in the standard medium; this concentration was similar to that of the luminal putrescine (Fig. 1a, d). Notably, the commercial organoid culture medium and supplements (Dulbecco's modified Eagle's medium [DMEM]/F12 supplemented with B27 and N2) also contained similar levels of putrescine. Thus, intestinal organoid growth may require exogenous putrescine. Treatment with *trans*-4-methylcyclohexylamine (MCHA; Fig. 2c), a spermidine synthase (SPDS) inhibitor, diminished the organoid growth (Fig. 2d). Thus, epithelial proliferation likely requires exogenous putrescine as a substrate for spermidine synthesis.

**Hypsination of eIF5A mediates the polyamine-dependent epithelial proliferation**. We subsequently explored the underlying mechanism by which polyamines regulate host epithelial proliferation. Ki67-positive proliferative CECs were increased in WT-strain-associated mice (Supplementary Fig. 3d). Nevertheless, mRNA expressions of *Mki67* (encoding Ki67) did not significantly differ between the WT- and SK930-strain-associated mice (Fig. 2e). Thus, polyamines may augment proliferation-related molecules via post-transcriptional regulation. Hyp-eIF5A, for which spermidine serves as a substrate, is involved in post-transcriptional regulation, including translation initiation, elongation, and termination[10–12]. Therefore, we thought that

MCHA-mediated inhibition of organoid growth may result from the reduction of hyp-eIF5A levels. Indeed, MCHA treatment decreased the hyp-eIF5A levels in the organoids, and exogenous spermidine rescued this decrease (Fig. 2f), indicating that the reduced hyp-eIF5A due to the inhibited spermidine synthesis arrested organoid growth. However, because the organoid growth was significantly suppressed, but hyp-eIF5A was only moderately reduced, MCHA may inhibit growth via other mechanisms in addition to reducing hyp-eIF5A.

We conducted organoid experiments to evaluate the difference in hyp-eIF5A levels in the CECs of two gnotobiotic mouse models. Colonisation by the WT strain, but not the SK930 strain, significantly increased the hyp-eIF5A levels in the CECs (Fig. 2g). To verify the contribution of hyp-eIF5A to epithelial proliferation, we treated colonic organoids with N1-monoguanyl 1,7-diaminoheptane (GC7)[46], which inhibits deoxyhypusine synthase (DHS) necessary for hyp-eIF5A synthesis (Fig. 2c). This treatment dose-dependently suppressed the organoid growth in conjunction with reducing the hyp-eIF5A levels (Fig. 2h, i), indicating that hyp-eIF5A deficiency suppressed organoid growth. *Lou* et al. reported that GC7 exerts little cytotoxicity up to 20 μM, whereas higher GC7 concentrations $(50 - 100$ μM) significantly inhibit hepatocellular carcinoma cell viability[47]. Therefore, we believe that the growth arrest of colonic organoids by 15 μM of GC7 did not result from the cytotoxicity of GC7. Furthermore, orally administering GC7 to SPF mice via their drinking water for 5 days markedly reduced the hyp-eIF5A levels and EdU-positivity in the CECs compared with those of the control mice who received regular tap water (Fig. 2j, k). Thus, GC7 suppressed CEC proliferation with decreased hyp-eIF5A levels both in vivo and in vitro.

Autophagy is closely related to hyp-eIF5A and is associated with proliferation and differentiation[48]. Autophagy is also activated in the crypts, which are proliferating regions, in CECs[49]. To assess whether putrescine produced by the intestinal microbiota increases autophagy activity, we detected LC3 and SQSTM1/p62 (p62) in CECs located 50 μm from the bottom of the crypt via immunohistochemical analysis. WT-strain-associated mice exhibited more LC3 dots than did the SK930-strain-associated mice (Fig. 3a). Additionally, the WT-strain-associated mice exhibited significantly more p62 dots and LC3-p62 double-positive dots than did the SK930-strain-associated mice (Fig. 3a, b). The LC3, p62, and LC3-p62 dots represent autophagosomal formation, sequestosomal formation (which is selectively degraded by autophagy), and autophagosome/autolysosome formation (incorporation of p62-positive sequestosomes into the autophagosome), respectively[50]. Thus, autophagy was activated in the CECs of WT-strain-associated mice compared with that in the CECs of SK930-strain-associated mice.

Hyp-eIF5A promotes efficient expression of mitochondrial oxidative phosphorylation (OXPHOS) proteins[26]. Therefore, we evaluated OXPHOS protein complexes I–V in the CECs of gnotobiotic mice. CI was significantly higher, and CII, CIV, and CV tended to be higher in SK930-strain-associated mice than in the WT-strain-associated mice, indicating that enhanced hyp-eIF5A did not promote OXPHOS in the WT-strain-associated mice (Fig. 3c). Crypt cells depend on glycolysis more than on OXPHOS for ATP production, whereas the dependence of enterocytes on OXPHOS to fulfil their energetic requirements increases with differentiation and maturation[51]. Therefore, these results were likely obtained because the WT-strain-associated mice exhibited more proliferating cells in the CEC crypts via polyamines derived from the WT strain than did the SK930-strain-associated mice. Thus, in this case, cell proliferation was likely more strongly influenced by the difference in energy metabolism in the CECs than by the hyp-eIF5a-OXPHOS axis.

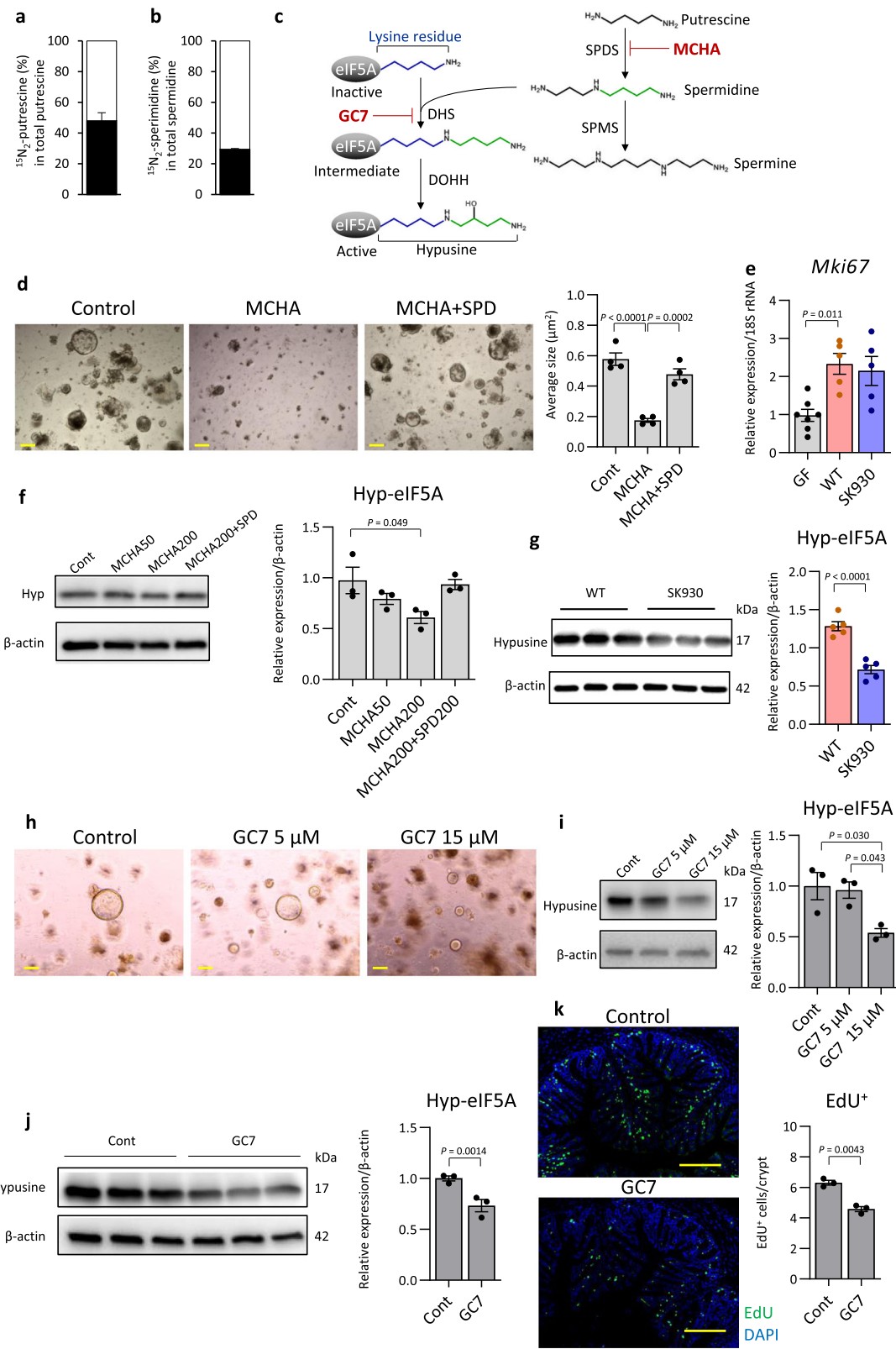

## Bacterium-derived polyamines regulate macrophage balance in the colon.

A previous study found that polyamines downregulate pro-inflammatory molecules, including nitric oxide (NO) and TNF, in LPS-stimulated macrophages and microglia[24,52]. In support of this view, ablation of ODC, a rate-limiting enzyme in polyamine synthesis, augments pro-inflammatory cytokines and chemokines in macrophages, reinforcing the M1 macrophage phenotype[25]. M2 macrophages, which play vital roles in tissue remodelling and resolution of inflammatory responses, are characterised by expression of arginase-1 (Arg1), which produces

**Fig. 2 Inhibition of hypusination of eIF5A arrests colonic organoid growth. a, b** Isotope ratio of putrescine and spermidine in colon organoids cultured with isotope-labelled $^{15}N_2$ putrescine were analysed via GC-MS. The black bar indicates the percent putrescine (PUT) or spermidine (SPD) of $^{15}N_2$ putrescine or $^{15}N_2$ spermidine enrichment in total corresponding polyamines, respectively ($n = 3$, independent cultures). **c** Polyamine synthesis pathway and eIF5A hypusination. Hypusine is formed by conjugation of the aminobutyl moiety (green) of spermidine to a lysine residue. MCHA and GC7 inhibit SPDS and DHS, respectively. SPMS: spermine synthase, DOHH: deoxyhypusine hydrogenase. **d** Representative microscopy images of organoid cultured with or without MCHA and spermidine (SPD). Scale bars: 200 μm. The average size of each organoid was quantified using Image J software ($n = 4$, independent cultures). **e** *Mki67* mRNA expression was analysed by q-PCR (GF: $n = 8$, WT: $n = 6$, SK930: $n = 8$, independent animals). Western blot analysis of hyp-eIF5A in **f** organoids treated with MCHA ($n = 3$) and in **g** CECs of gnotobiotic mice ($n = 5$). **h** Representative microscopy images of organoids cultured with or without GC7. Scale bars: 200 μm. Reproducibility of results was confirmed by performing two independent experiments. **i** Western blot analysis of hyp-eIF5A in organoids treated with MCHA ($n = 3$, independent cultures). **j** Western blot analysis of hyp-eIF5A in CECs of SPF mice with GC7 treatment ($n = 3$, independent animals). **k** Representative microscopic images of EdU assay and quantification of EdU-positive cells/crypt in control or GC7 treated mice. EdU-positive cells (green). Nuclei are counterstained with DAPI (blue). Scale bars: 100 μm. Three randomly selected areas were examined per slide ($n = 3$, independent animals). All data shown represent the mean ± SEM. Statistical significance was calculated using **d, f** the one-way ANOVA followed by Tukey's post-hoc test, **e, i** Welch's ANOVA followed by Dunnett's T3 post-hoc test and **g, j, k** Student's *t*-test (two-sided).

ornithine[53]. Therefore, we hypothesised that commensal-derived polyamines might significantly affect the M1/M2 macrophage balance. Under physiological conditions, two major macrophage/monocyte populations exist in the colonic lamina propria (cLP): $CX_3CR1^{low}Ly6C^+$ cells and $CX_3CR1^{high}Ly6C^-$ cells, which display inflammatory and anti-inflammatory properties, respectively[29]. WT strain colonisation was increased in the $CX_3CR1^{high}Ly6C^-$ subset in the cLP but not in the spleen (Fig. 4a–c and Supplementary Fig. 5a, b). Additionally, an $Arg1^+$ M2-like subset was significantly increased in the WT-strain-associated mice compared with that in the GF mice. Conversely, association with the SK930 strain failed to affect the $CX_3CR1^{high}Ly6C^-$ subset and significantly increased the frequency of $NOS2^+$ M1-like macrophages in SK930-strain-associated mice, thus augmenting the M1/M2 ratio in the cLP (Fig. 4d–g).

Bacterial putrescine exerts a localised effect in regulating the macrophage balance. To determine whether exogenous polyamines directly regulate macrophage polarisation, bone marrow-derived macrophages (BMDMs) were cultured in medium with or without putrescine. Similar to the results of the WT- and SK930-strain-associated mice, exogenous putrescine upregulated $CX_3CR1$ expression and downregulated Ly6C expression in the BMDMs (Fig. 4h, i). To further confirm that macrophages take up exogenous polyamines, we incubated BMDMs in a complete-RPMI medium supplemented with 100 μM $^{15}N_2$-putrescine for 24 h. The intracellular isotope-labelled polyamines, $^{15}N_2$-putrescine and $^{15}N_2$-spermidine, occupied 81.5% and 60.6%, respectively, of the corresponding polyamines in the BMDMs (Fig. 4j, k), demonstrating that exogenous putrescine was actively uptaken and converted to spermidine in the BMDMs.

A recent study revealed that polyamine biosynthesis causes eIF5A hypusination in macrophages, and hyp-eIF5A modulates expression of OXPHOS-related mitochondrial proteins to induce alternative macrophage activation[26]. To assess the effect of exogenous putrescine on hyp-eIF5A and OXPHOS, we treated bone marrow cells with macrophage colony-stimulating factor and the presence or absence of DFMO and putrescine. Hypusination of eIF5A was upregulated in macrophages stimulated with IL-4, but not in those stimulated with LPS + IFN-γ, which is similar to the results of previous studies (Supplementary Fig. 6)[26]. DFMO treatment decreased hyp-eIF5A in BMDMs, and exogenous putrescine rescued this DFMO-induced reduction in hyp-eIF5A. Meanwhile, exogenous putrescine treatment without DFMO slightly increased the hyp-eIF5A (Supplementary Fig. 6). We further analysed whether polyamines cause metabolic reprograming in BMDMs. The expressions of OXPHOS complex proteins CI–CV were assessed via western blotting. CI, CII and CIV were upregulated in macrophages stimulated with IL-4, but not in those stimulated with LPS + IFN-γ, consistent with the previous studies (Supplementary Fig. 6)[26]. Additionally, CI, CII and CIV expressions were decreased in DFMO-treated macrophages, and exogenous putrescine restored this suppression. OXPHOS complex protein expression in macrophages treated only with putrescine resembled that in macrophages stimulated with IL-4 (Supplementary Fig. 6). Thus, exogenous putrescine induced metabolic reprogramming similar to that of IL-4 stimulation.

**Bacterium-derived polyamines mitigate DSS-induced colitis.** Vigorous epithelial proliferation is a fundamental mucosal barrier function. $CX_3CR1^{high}Ly6C^-$ macrophages contribute to suppressing inflammatory responses in the intestine. Therefore, we explored whether bacterial polyamines may substantially contribute to maintaining intestinal immune homoeostasis. We administered DSS in the drinking water to F1 gnotobiotic mice to induce experimental colitis. The number of *E. coli* during DSS-induced colitis did not differ between the WT- and SK930-associated mice (Supplementary Fig. 7a); however, WT-strain-associated mice were more resistant to the DSS-induced colitis than were the SK930-strain-associated mice, as evidenced by the significant amelioration of disease symptoms and increased survival rates in the WT-strain-associated mice (Fig. 5a–c). The WT-strain-associated mice also exhibited alleviated colonic thickening, histological scores and faecal lipocalin-2 levels (Fig. 5d–f). Flow cytometry showed an accumulation of $CD11b^+F4/80^+$ macrophages/monocytes (Supplementary Fig. 7b, c), especially the $CX_3CR1^{low}Ly6C^+$ and $NOS2^+$ M1-like subsets, in the cLPs of the SK930-strain-associated mice (Fig. 5g–i and Supplementary Fig. 7d–f). Conversely, colonisation by the WT strain decreased the M1/M2 macrophage balance under inflammatory conditions (Fig. 5j). Thus, bacterium-derived polyamines play anti-inflammatory roles in the intestinal immune system. Interestingly, a recent study demonstrated that spermine oxidase deficiency reduced colonic spermidine levels, which exacerbated lethality and mucosal inflammation, in a DSS-induced colitis model[54]. This observation supports the importance of polyamine metabolism in maintaining intestinal immune homoeostasis. Of note, a recent work exhibited that gastrointestinal pathobionts such as *Helicobacter pylori* and *Citrobacter rodentium* induce accumulation of the intracellular hyp-eIF5A level in macrophages by upregulating DHS[55]. DHS-deficient macrophages are defective in antibacterial response to these pathobionts, suggesting that polyamine-dependent formation of hyp-eIF5A is essential to prevent bacterial infection on the gastrointestinal mucosa.

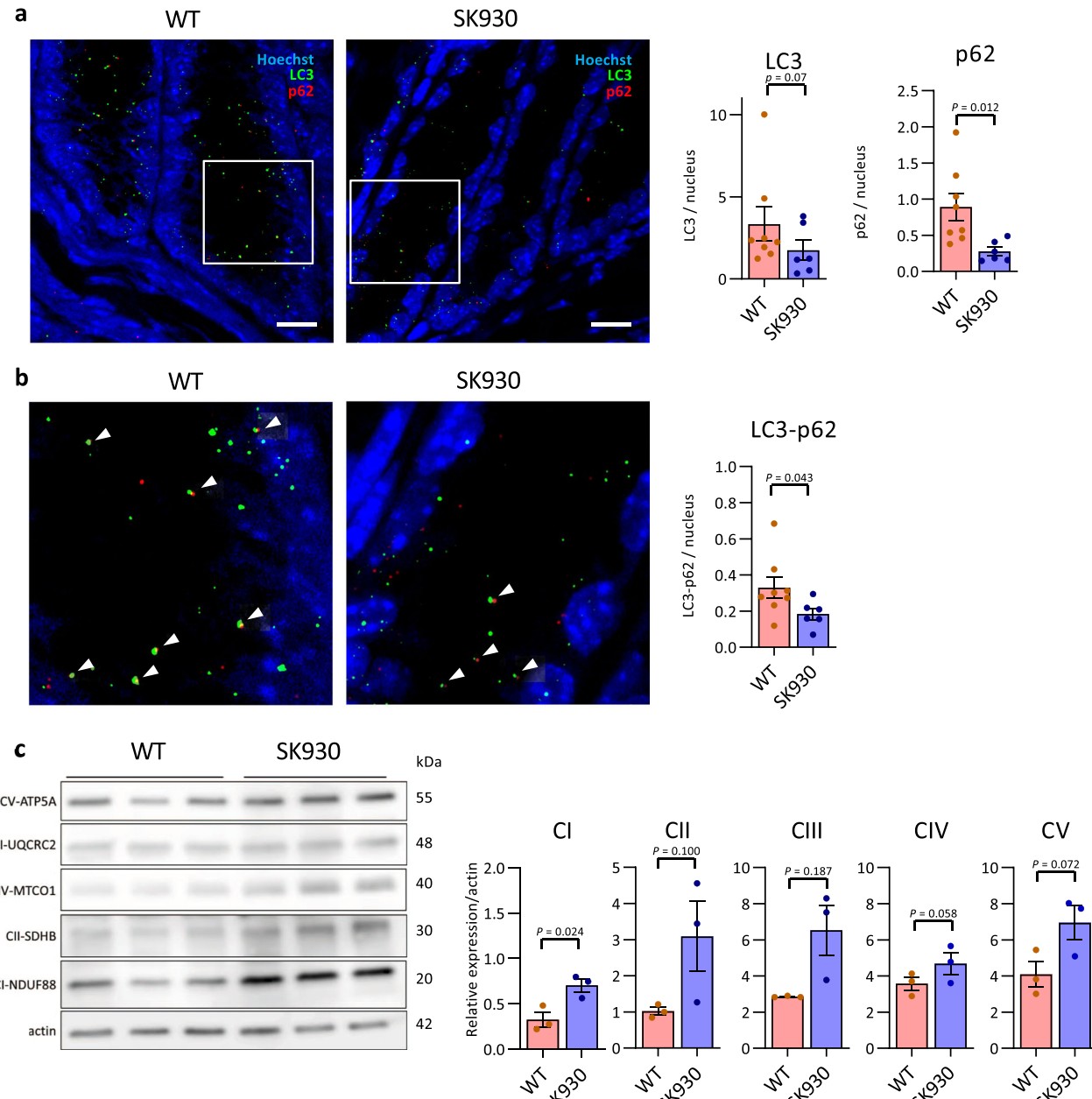

**Fig. 3 Bacterial putrescine activates autophagy and OXPHOS in CECs. a, b** Representative microscopic images of LC3 and p62 and its enlarged view. Green and red dots indicate LC3 and p62, respectively. Nuclei are stained with Hoechst (blue). The white arrowhead indicates LC3-p62 double-positive dots. Scale bars: 10 μm. Three randomly selected areas were examined per slide (WT: $n = 8$, SK930: $n = 6$, independent animals). **c** Western blot analysis of OXPHOS complex I-V (CI-CV) in CECs of gnotobiotic mice ($n = 3$, independent animals). All data shown represent the mean ± SEM. Statistical significance was calculated using the Welch's $t$-test (**a**, p62 and **c**) (two-sided), Mann–Whitney test with (**a**, LC3 and **b**, LC3-p62) (two-sided).

*Carriche* et al. revealed that spermidine promoted differentiation into CD4+Foxp3+ T cells in vitro[56], and oral administration of spermidine increased CD4+Foxp3+ T cells in the small intestines and cLP in vivo. However, the CD4+Foxp3+ T cells did not significantly differ between the cLPs of the GF, WT- and SK930-strain-associated mice in our experiments (Supplementary Fig. 5c). Therefore, CD4+Foxp3+ cells were not involved in the results of the DSS-induced colitis experiment in this study.

To our knowledge, this study provides the first direct evidence that host cells incorporate polyamines from the colonic lumen, which eventually increase intracellular polyamine levels to regulate the mucosal epithelium and immunity. Recent studies

exhibited that *ODC* expression was downregulated in the inflamed region compared with that in the uninflamed region of the colonic mucosa in patients with Crohn's disease[57,58]. Thus, decreased endogenous polyamine synthesis might be involved in the development and/or exacerbation of inflammatory bowel disease (IBD). Therefore, our study may suggest an alternative therapeutic strategy for IBD and metabolic syndrome by manipulating microbial polyamine synthesis using prebiotics, probiotics, or synbiotics (e.g., arginine and *Bifidobacterium* spp.)[30]. Notably, polyamine biosynthesis is highly upregulated in cancerous cells, and selective inhibition of ODC is effective in treating colon cancer[59]. This suggests the pathological relevance

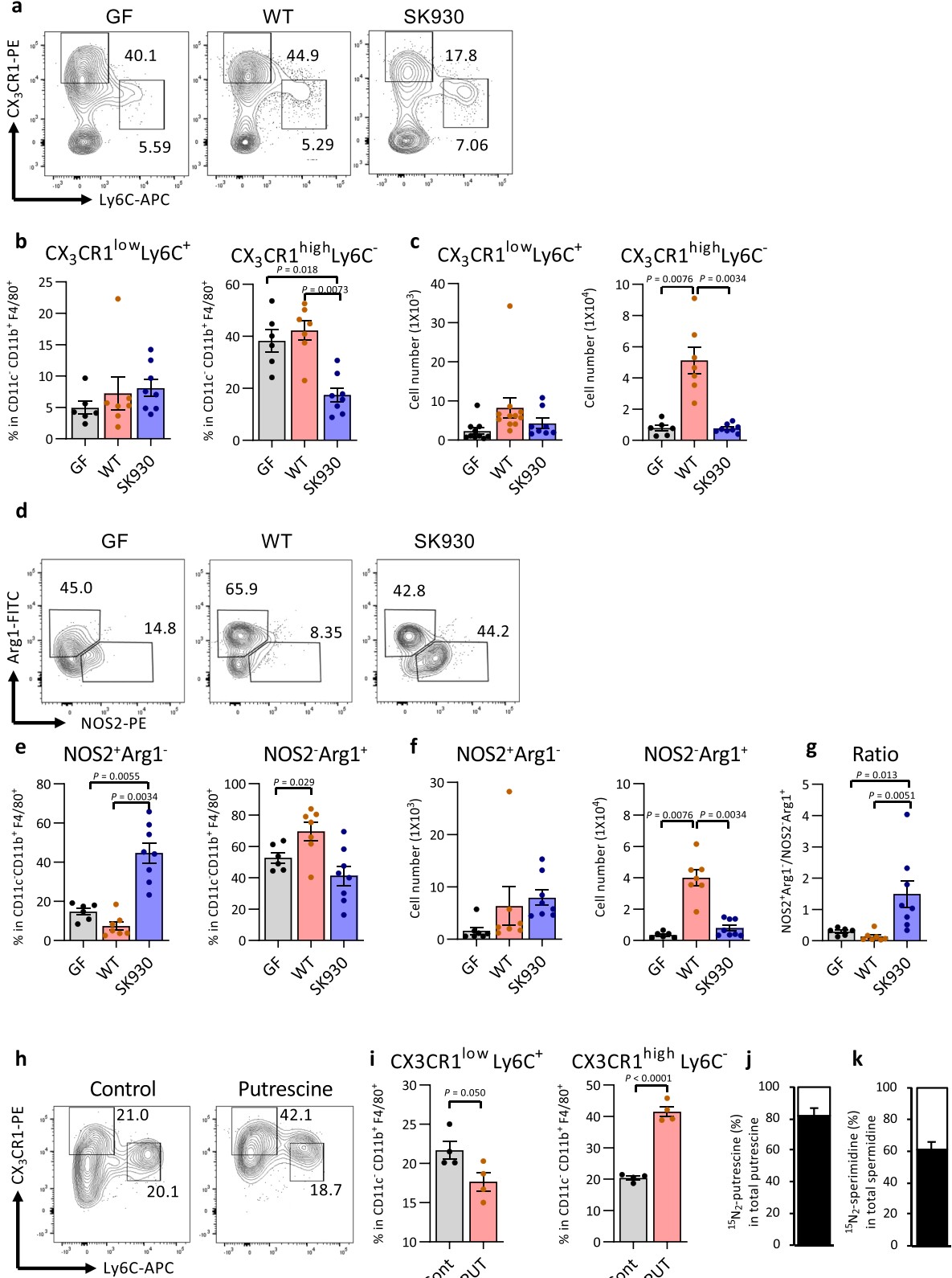

of polyamines produced by commensals in colorectal carcinogenesis, although further investigation is required to confirm this notion.

In conclusion, low-molecular-weight metabolites produced by the intestinal microbiome are absorbed from the colonic lumen and carried into the systemic circulation. Some of these metabolites play vital roles in the health and diseases of host animals. Accumulating evidence shows that bioactive metabolites, such as SCFAs, act directly on host cells to exert physiological functions. Additionally, host cells may further metabolise certain

**Fig. 4 Bacterial putrescine regulates the development of macrophage subsets.** $CX_3CR1$ and Ly6C expression were analysed in $CD45^+CD11b^+F4/80^+$ cells via flow cytometry. Representative flow cytometry images (**a**), frequency (**b**), number (**c**) of $CX_3CR1^{low}Ly6C^+$ monocyte/macrophage and $CX_3CR1^{high}Ly6C^-$ macrophages and representative flow cytometry images (**d**), frequency (**e**), number (**f**) of $NOS2^+Arg1^-$ and $NOS2^-Arg1^+$ macrophages and the ratio of the former to the latter subset (**g**) in cLP of gnotobiotic mice colonised with WT ($n = 7$) or SK930 ($n = 8$) strain and GF mice ($n = 6$). $CX_3CR1$ and Ly6C expression in $CD45^+CD11c^-CD11b^+F4/80^+$cells from BMDMs cultured in medium with or without putrescine ($n = 4$). Representative flow cytometry images (**h**) and frequency (**i**). **j, k** Isotope ratio of putrescine and spermidine in BMDMs cultured with isotope-labelled $^{15}N_2$-putrescine were analysed via GC-MS. The black bar indicates the percent putrescine (PUT) or spermidine (SPD) of $^{15}N_2$-putrescine or $^{15}N_2$-spermidine enrichment in total corresponding polyamines, respectively ($n = 3$, independent cultures). All data shown represent the mean ± SEM. Kruskal-Wallis followed by Steel-Dwass post hoc test (**b-g**) and Student's $t$-test (**i**) (two-sided).

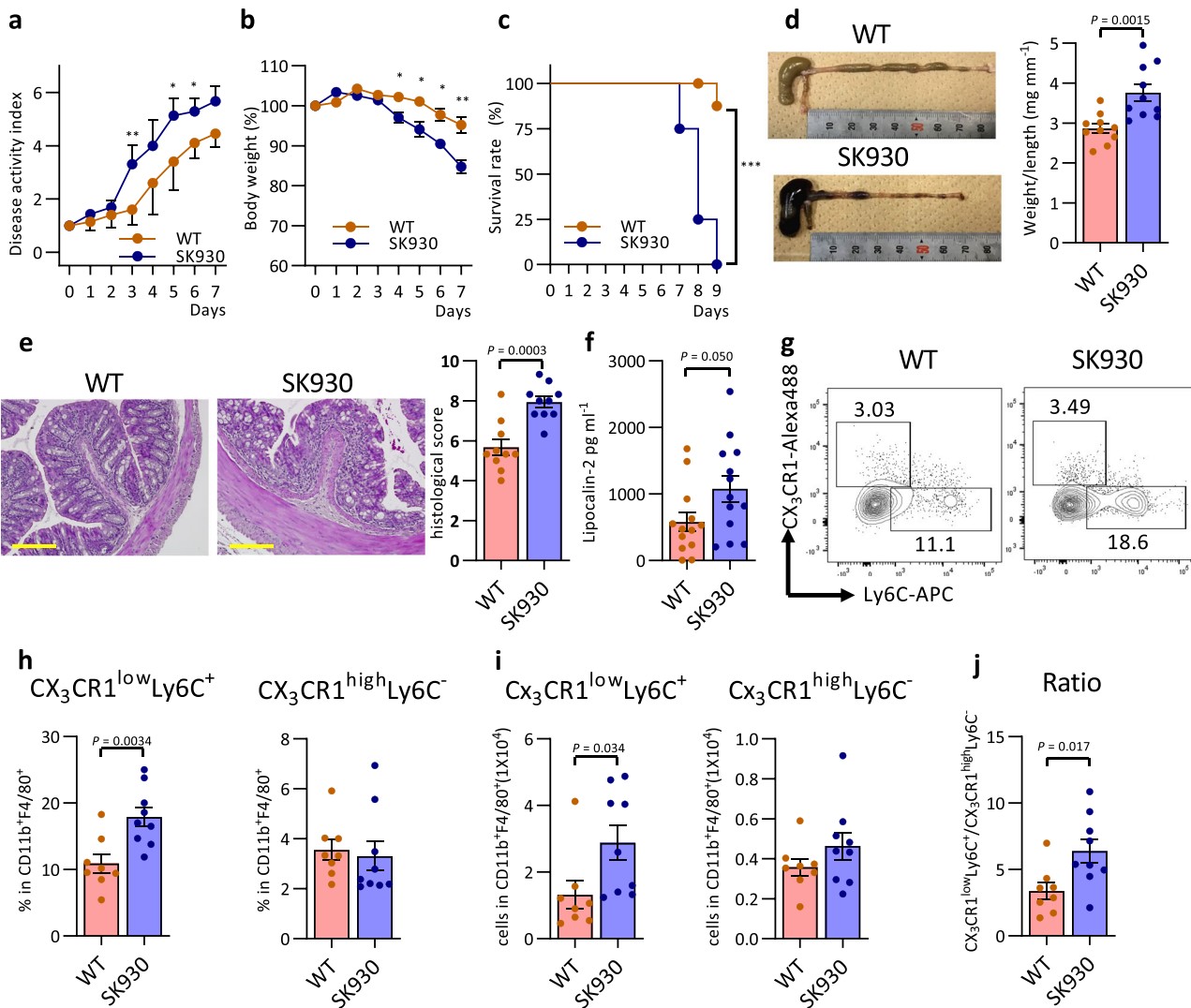

**Fig. 5 Bacterial putrescine ameliorates DSS-induced colitis.** 2% DSS in the drinking water to the F1 gnotobiotic mice colonised WT or SK930 strain to induce experimental colitis for six days, then regular drinking water thereafter. **a–c**: Red and blue indicate WT-strain-associated and SK930-strain-associated mice. **a** DAI score. **b** Body weight changes were measured daily and calculated as the percentage change from day 0. **c** Survival rate in mice after DSS administration. The survival curve was calculated using the Kaplan-Meier method and statistical significance was calculated using the log rank test (WT: $n = 10$, SK930: $n = 8$). **d–j** Mice were analysed on day 5 after starting 2% DSS ($n = 10$). **d** Representative image of the colons and values of weight/length of the colon. **e** Haematoxylin and eosin-stained images of representative histopathologic changes in the WT and SK930 strain-associated mice. Scale bar: 100 µm. Mucosal damage was estimated using a histological scoring system. **f** Faecal lipocalin-2 concentrations in faeces at day 5 after starting 2% DSS. Expression of $CX_3CR1$ and Ly6C were analysed in $CD45^+CD11b^+F4/80^+$ cells via flow cytometry. Representative flow cytometry images (**g**), and frequency (**h**) and number (**i**) of $CX_3CR1^{low}Ly6C^+$ monocyte/macrophages and $CX_3CR1^{high}Ly6C^-$ macrophages and the ratio of the former to the latter subset (**j**) in the cLP of gnotobiotic mice colonised with WT or SK930 strain (WT: $n = 8$, SK930: $n = 9$). All data shown represent the mean ± SEM. Statistical significance was calculated using the Welch's $t$-test (**e, h, i**) (two-sided) with Bonferroni adjustments (**a**), Mann–Whitney test with Bonferroni adjustments (**b**), Student's $t$-test (**d, f, j**) (two-sided) (*$P < 0.05$, **$P < 0.01$, ***$P < 0.001$).

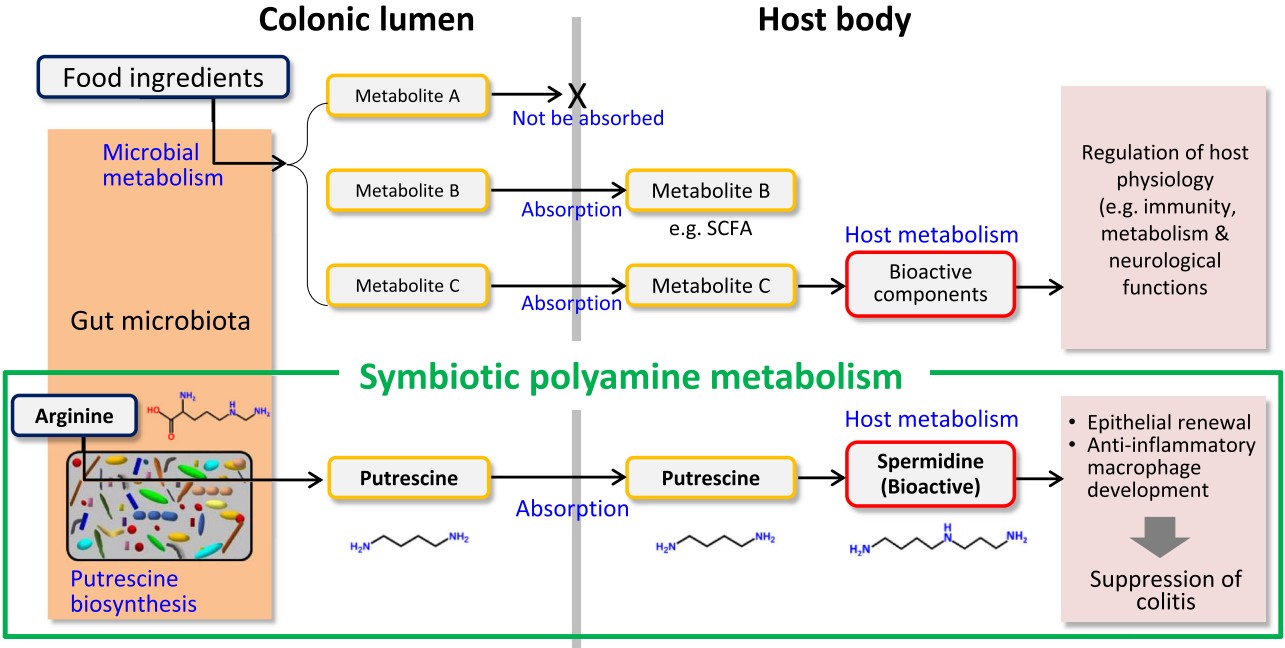

**Fig. 6 Symbiotic metabolism.** Low-molecular metabolites produced by the intestinal microbiota remain in the lumen (Metabolite A) or are absorbed from the intestinal lumen to systemic circulation (Metabolite B and C). Metabolite B, which are represented by short-chain fatty acids, directory act on host cells to exert physiological activities. Meanwhile, a part of inactive metabolites may be further converted to bioactive substances via host metabolic pathways after absorption (Metabolite C). The entities of such symbiotic metabolism, however, remain obscure. Our study using gnotobiotic mice demonstrated that microbial putrescine is uptaken by the colonic epithelial cells to produce bioactive spermidine that accelerates epithelial renewal and increases the abundance of anti-inflammatory macrophages in the colon. Thus, microbial putrescine serves as a source (precursor) of bioactive substances in the host animals. This observation provides evidence for symbiotic metabolism contributing to the maintenance of intestinal homoeostasis.

microbial products to generate bioactive materials that potentially impact the host physiology (Fig. 6). Such biosynthetic pathways composed of both microbes and host cells could be termed symbiotic metabolism. Thus, our study revealed that symbiotic polyamine metabolism significantly contributes to fine-tuning the intestinal mucosal epithelium and immunity.

## Methods
**Bacterial strains.** *Escherichia coli* MG1655 was used as a wild-type (WT) strain. A triple mutant SK930 harbouring the deletion of putrescine synthesis genes encoding agmatine ureohydrolase (*speB*) and ornithine decarboxylase (*speC* and *speF*), was used as the knockout strain[30].

**Animal experiments.** All mice were purchased from Clea Japan, Inc. GF (IQI/Jic) mice were purchased and bred at Kyodo Milk Industry Co., Ltd. Faecal microbiome transplantation experiment: Mice were housed in plastic cages under a 12-hour light/dark cycle at $25 \pm 2\,^{\circ}\mathrm{C}$ with $50\% \pm 10\%$ humidity. Mice were provided with sterilised (for GF) or non-sterilised (for exGF and SPF) water and sterilised (for GF, exGF and SPF) CE-2 pellet chow (Clea Japan, Inc.). To perform faecal microbiome transplantation, eleven-week-old GF male mice were inoculated in the stomach via a catheter with a suspension of the faeces obtained from the SPF mice. The exGF and SPF mice were then analysed for metabolomic analysis of the caecal contents and CECs at 3, 6, 9, and 23 days after inoculation. Gnotobiotic mouse experiment: Mice were housed in flexible film plastic isolators with sterilised bedding and provided with sterilised water and sterilised low-polyamine pellet chow (modified AIN-93M formula with soybean oil in substitution of corn oil; Oriental Yeast Co., Ltd.) ad libitum.   Eight-week-old female IQI/jic GF mother mice (F0) were orally administered phosphate-buffered saline containing $1 \times 10^8$ colony-forming units (CFUs) of the viable WT or SK930 strains. Next, each strain was transmitted and colonised in the F1 generation via maternal-infant transmission. The F1 male mice were used in most of experiments, except the experiments in Fig. 5a–c where both female and male mice were used. C57BL/6 J male mice were used in the in vitro experiment with organoids and BMDMs culture. The Kyodo Milk Animal Use Committee (Permit Number: 2018-02) approved the protocols, which were performed were in accordance with the Guide for the Care and Use of Laboratory Animals, published by the National Academies Press.

**CE TOF-MS.** Intestinal epithelium samples were isolated and disrupted with 3.0-mm Zirconia Beads (Biomedical Science) in methanol by vigorous shaking (1,500 rpm for 10 min) using Shake Master (Biomedical Science). Cellular metabolites were extracted using the methanol:chloroform:water extraction protocol. Time-of-flight mass spectrometry (CE-TOFMS)–based metabolome experiments were performed using the Agilent CE System, the Agilent G3250AA LC/MSD TOF System, the Agilent 1100 Series Binary HPLC Pump, the G1603A Agilent CE-MS adapter and the G1607A Agilent CE-ESI-MS Sprayer Kit (Agilent Technologies, USA)[60]. In-house software (MasterHands) was used for data processing, quantification and peak annotation[61]. Heatmaps were drawn with normalised z-score using R software (ver. 3.5.3).

**Cultures of bacteria.** To determine the polyamine concentrations, each of the strains were grown under anaerobic conditions at $37\,^{\circ}\mathrm{C}$ for 24 h in LB medium (BD Biosciences, USA) containing 2 mM L-arginine, $1.5\,\mathrm{g\,L^{-1}}$ D-glucose, and $0.5\,\mathrm{g\,L^{-1}}$ L-cysteine-hydrochloride in screw-top test tubes with butyl rubber inner plugs. The gas-phase portions in screw-top test tubes were replaced with $N_2/CO_2$ (80:20, v/v) before autoclaving ($115\,^{\circ}\mathrm{C}$, 15 min). *E. coli* precultures were harvested after 24 h and inoculated at a final $OD_{600}$ of $1.0 \times 10^{-2}$. To determine the growth curve and the amount of LPS, each strain was grown under aerobic conditions at $37\,^{\circ}\mathrm{C}$ in LB medium. The growth curve was measured for up to 24 h using an absorptiometer with $OD_{600}$. LPS was extracted when $OD_{600}$ reached 0.3 by hot phenol-water method[62], and quantified using ToxinSensor Chromogenic LAL Endotoxin Assay Kit® (GenScript Inc., USA) following the manufacturer's protocol.

**Measurement of polyamine concentrations.** To prepare the polyamine extract, faeces were collected and stored at $-80\,^{\circ}\mathrm{C}$ until use. Frozen faeces (approximately 50 mg) were diluted 10-fold with Dulbecco's phosphate-buffered saline (D-PBS; Thermo Fisher Scientific, Inc., USA) and extracted three times by intense mixing for 1 min, then left to stand for 5 min in an icebox. The upper aqueous portion was collected via centrifugation ($12,000 \times g$ for 10 min at $4\,^{\circ}\mathrm{C}$). Eight-week-old GF or gnotobiotic mice were sacrificed, and their colons were opened using a longitudinal incision. The colonic tissue was washed twice in cold Hank's balanced salt solution (HBSS) and incubated with HBSS containing 30 mM EDTA and 1 mM dithio-threitol at $37\,^{\circ}\mathrm{C}$ and 5% $CO_2$ for epithelial detachment. After 20 min, the colons were washed with HBSS, and epithelial cells were detached using curbing needles in cold HBSS. Detached epithelial cells were collected via centrifugation ($500\,g$ for 5 min at $4\,^{\circ}\mathrm{C}$) and washed twice with HBSS. Some of the epithelial cells were soaked in RNAlater (Thermo Fisher Scientific) for RNA extraction. The remaining

epithelial cells were added to RIPA buffer (Thermo Fisher Scientific) supplemented with protease inhibitor cocktail (Thermo Fisher Scientific) and homogenised by passing through a 26 G needle 20 times. The upper aqueous portion was collected via centrifugation (15,000 g for 10 min at 4 °C). The aqueous portion was also used to analyse protein expression. The organoids were dissociated using GentleMax (Miltenyi Biotec, Germany) and washed with HBSS. Extraction was then performed in the same manner as that for the CECs. The polyamine concentrations in each extraction were derivatised using ethyl chloroformate and trifluoroacetic acid anhydride with 1,6-diamino hexane as an internal standard[63,64]. The derivatised samples were injected with a 1.5:1 split into a GCMS-QP2010 gas chromatography coupled with a mass spectrometer detector (GC-MS; GCMS-QP2010, Shimadzu, Co., Japan). Helium was used as a carrier gas. Samples were analysed using a ZB-5 capillary column (60 m × 0.25 mm, 0.25-µm film thickness, Shimadzu). The injector and source temperatures were 260 °C and 150 °C, respectively. The GC oven was programmed as follows: the starting temperature of 140 °C was increased to 190 °C at a rate of 8 °C/min, increased to 320 °C at a rate of 20 °C/min, and finally held for 5 min. Quantification was performed using selected ion monitoring. The ions were monitored at m/z 355 for putrescine, m/z 383 for 1,6-diamino hexane, m/z 480 for spermidine and m/z 609 for spermine. The extraction and derivatization rates were standardised using 1,6-diamino hexane and quantified using the corresponding external calibration curves[63,64].

**Measurement of SCFAs.** Faecal extracts were obtained using the same method used to measure the polyamine concentrations. Faecal extracts were spiked with 2-ethyl butyric acid (2-EB; Sigma-Aldrich, USA) as an internal standard and deproteinised with 10% (v/v) of 20% (w/v) 5-sulfosalicylic acid (Sigma-Aldrich) solution. The deproteinised extracts were acidified with 37% (w/v) hydrochloric acid, and organic acids were extracted with diethyl ether by vortexing for 15 min. The samples were centrifuged for 5 min at 15,000 × g to separate into two phases; the upper organic layer was collected in a glass vial. After adding N-tert-butyldimethylsilyl-N-methyltrifluoroacetamide (MTBSTFA; Sigma-Aldrich) as a derivatization reagent, the samples were incubated for 24 h at room temperature in the dark. The derivatised samples were injected with a 5:1 split into GCMS-QP2010 (Shimadzu). Helium was used as a carrier gas. Samples were analysed using a ZB-5 capillary column (60 m × 0.25 mm, 0.25-µm film thickness, Shimadzu). The injector and source temperatures were 250 °C and 200 °C, respectively. The GC oven was programmed as follows: the starting temperature of 55 °C was increased to 70 °C at a rate of 10 °C/min, increased to 280 °C at a rate of 20 °C/min, and finally held for 3 min. Quantification was performed using selected ion monitoring. The ions were monitored at m/z 117 for acetic acid, m/z 131 for propionic acid, m/z 145 for butyric acid and isobutyric acid, m/z 159 for valeric acid and isovaleric acid, m/z 173 for 2-EB, m/z 261 for lactic acid, and m/z 289 for succinic acid. The extraction and derivatization rates were standardised using 2-EB acid and quantified using the corresponding external calibration curves[65,66].

**Measurment of proliferation.** For the EdU assay, mice were injected intraperitoneally with 50 µg 5-ethynyl-2′-deoxyuridine (EdU; Kanto Chemical, Co. Inc., Japan) 3 h before sacrifice. Colonic tissue samples were obtained from 1 cm above the anus and fixed in 10% neutral buffered formalin solution Mildform 10 N (Wako Pure Chemical Industries, Ltd., Japan). After fixation, samples were processed stepwise through methanol, xylene, and paraffin. The samples were embedded in paraffin and cut into 3-µm sections. The sections were deparaffinised and rehydrated, and EdU was detected using Click-iT® EdU Imaging Kits (Thermo Fisher Scientific) following the manufacturer's protocol. For the Ki67 immunostaining, deparaffinised sections were subjected to antigen retrieval using citrate buffer (DAKO S-1700; DAKO, Co., Denmark) following the manufacturer's protocol. The sections were washed with D-PBS and incubated with blocking by 2% goat serum in 0.5% blocking reagent (Perkin Elmer, Co., USA) for 30 min at room temperature. After removing the excess blocking reagent, the sections were incubated overnight at 4 °C with anti-Ki67 antibody (NCL-Ki67p, Novocastra, UK; 1:1000) in blocking buffer. The next day, the sections were washed with D-PBS with 0.05% Tween, twice for 5 min, then incubated for 2 h with goat anti-rabbit-IgG-488 antibody (A-11034, Thermo Fisher Scientific; 1:400) in blocking buffer. After being washed, the slides were mounted with VECTASHIELD (Vector Laboratories, Inc., USA), including DAPI. Three randomly selected areas were measured per slide. For the LC3 and p62 immunostaining, deparaffinised sections were subjected to antigen retrieval by heating using the microwave for 15 min in 0.01 M citrate buffer (pH = 6.0). The sections were washed with D-PBS and incubated with blocking by 10% goat serum for 30 min at room temperature. After removing the excess blocking reagent, the sections were incubated overnight at 4 °C with anti-LC3 antibody (PM036; Medical and Biological Laboratories Co., Ltd.; Japan; 1:1000) and anti-p62 antibody C-terminal (PM066; Medical and Biological Laboratories; 1:200) in blocking buffer. The next day, the sections were washed with D-PBS, three times for 5 min, then incubated for 1 h at room temperature with goat anti-rabbit-IgG antibody conjugated Alexa Fluor 488 (A-11008; Thermo Fisher Scientific; 1:500) and biotinylated goat anti-guinea pig-IgG antibody (BA-7000; Vector Laboratories, Inc.; California, USA; 1:200) in blocking buffer. The sections were washed with D-PBS, three times for 5 min, and then the sections were incubated for 30 min at room temperature with streptavidin conjugate Alexa Fluor™ 555 (S21381; Thermo Fisher Scientific; 1:500) and Hoechst 33342 solution

(H342; DOJINDO; Japan; 1:1000) in D-PBS. After being washed three times, the slides were mounted with ProLong™ Diamond Antifade Mountant (Thermo Fisher Scientific). Three or four randomly selected areas were measured per slide.

**Quantitative RT-PCR.** Total RNA was extracted from the CECs and BMDMs using NucleoSpin® RNA (TaKaRa Biotechnology Co., Ltd., Japan) following the manufacturer's protocols. The total RNA was reverse-transcribed to cDNA using the PrimeScript™ RT Reagent Kit (TaKaRa Biotechnology). The synthesised cDNA was used as a template for the quantitative RT-PCR experiments using the StepOnePlus™ real-time PCR system (Thermo Fisher Scientific) with TB Green Premix Ex Taq II (Tli RNaseH Plus) (TaKaRa Biotechnology). Relative expression levels were calculated using the ΔΔCt method with 18 S rRNA as the reference genes. Primer sequences were described in Supplementary Table 2.

**Western blotting.** Rabbit anti-hypusine antibody (ABS1064, Merck Millipore, USA; 1:5000), mouse anti-β-actin antibody (A1978, Sigma-Aldrich; 1:2500) and mouse anti-actin antibody (017–2455, Fujifilm Wako Pure Chemical Corporation; 1:5000) were used for detection of hypusine protein expression and as a loading control. Total OXPHOS Rodent WB Antibody Cocktail (ab110413, Abcam, Cambridge, UK; 1:1000) was used for detection of mitochondrial OXPHOS-complex expression and as a loading control. Protein denatured at 95 °C for 5 min were separated via SDS-PAGE on the 4–20% Mini-PROTEAN® TGX™ Gels (Bio-Lad Laboratories, Inc., USA), then electrophoretically transferred to Immuno-Blot® poly vinylidene difluoride membrane (Bio-Lad Laboratories). Protein denaturation was not performed to detect total OXPHOS. Protein was detected using horseradish peroxidase-labelled secondary antibodies and the enhanced chemiluminescence system. Band intensity was quantified using ImageJ software (version 1.47 v, National Institutes of Health, USA).

**Inhibition of DHS in vivo.** Seven-week-old SPF IQI/Jic mice were administrated GC7 (Santa Cruz Biotechnology Inc., CA, USA) of 500 µM in drinking water for 5 days. Mice were injected intraperitoneally with 50 µg EdU (Kanto Chemical) 3 h before sacrifice. Western blotting and EdU assay mentioned above.

**Colonic organoid cultures.** Colonic crypts were isolated from the C57BL/6 J mouse colons. The colonic tissue was washed and incubated in 10 ml of isolation buffer (43.4 mM L-sucrose·54.9 mM D-sorbitol·96.2 mM NaCl·5.6 mM Na2HPO4/12H2O·1.6 mM KCl·8.0 mM KH2PO4 in deionised water sterilised through a 0.22-µm filter) containing 2 mM EDTA and 0.5 mM DTT for 75 min. The colonic tissue was then rinsed with 5 ml isolation buffer twice and vigorously shaken by hand in 10 ml isolation buffer to release the crypts[67]. The isolated crypts were mixed with Matrigel (BD Biosciences) and transferred to 48-well (for the putrescine tracing experiment and inhibition of spermidine synthesis) or 24-well plates (for inhibition of DHS). After polymerising the Matrigel, standard medium[45] was overlaid on the gel in each well. For the putrescine tracing experiment, $^{15}N_2$-putrescine 2·HCl with atomic purity >98% (Sigma-Aldrich) was solubilised in saline and neutralised to pH 7.0. The organoids were cultured with standard medium for 5 days, and then the organoids were cultured with standard medium added with stable isotope-labelled putrescine for 2 days. Isotope enrichment in organoids lysed with RIPA buffer was analysed via GC-MS. Inhibition of spermidine synthesis: MCHA (Tokyo Chemical Industry Co., Ltd, Japan)and spermidine were added to the organoid medium from day 1, and organoids were grown for 5 days. The medium was replaced every 2 days. Inhibition of DHS: GC7 was added to the culture medium from the 4th to the 6th day after initial culturing the organoids for 3 days without GC7. The average size of each organoid was quantified using ImageJ software. The protein was extracted RIPA buffer (Thermo Fisher Scientific) supplemented with a protease inhibitor cocktail (Thermo Fisher Scientific) by sonication.

**Preparation of immune cells.** Colonic tissues were treated with HBSS containing 1 mM dithiothreitol, 20 mM ethylenediaminetetraacetic acid (EDTA) and 12.5 mM HEPES at 37 °C for 30 min on a stirrer. The tissues were then minced and dissociated in RPMI-1640 containing 0.5 mg ml⁻¹ collagenase (Wako), 0.125 mg ml⁻¹ DNase I, 2% new-born calf serum, 100 U ml⁻¹ penicillin, 100 µg ml⁻¹ streptomycin and 20 mM HEPES (2 R media) at 37 °C for 30 min to obtain single-cell suspensions. Single splenic leucocyte suspensions were prepared by mechanically disrupting the tissues through 100-µm nylon-mesh cell strainers (BD Biosciences) in 2 R media. These preparations were then subjected to flow cytometry analysis using the LSR II flow cytometer (BD Biosciences).

**Flow cytometry.** Leucocytes were incubated with anti-CD16/CD32 antibodie (AT-10, BioLegend, USA) to block the Fc receptors. Macrophages from cLP and BMDMs were surface stained with BV 510-conjugated anti-CD45 antibody (30-F11, BioLegend), FITC-conjugated anti-CD11b antibody (M1/70, Thermo Fisher Scientific), BUV395-conjugated anti-CD11c antibody (HL3, BD Biosciences, PerCP/Cy5.5-conjugated anti-F4/80 antiboy (BM8, BioLegend), PE-conjugated anti-CX₃CR1 antibody (SA011F11, Biolegend) and APC-conjugated anti-Ly6C antibody (AL-21, BD Biosciences). Macrophages from spleen were surface stained with BV 510-conjugated anti-CD45 antibody (30-F11, BioLegend), PE-conjugated

anti-CD11b antibody (M1/70, Thermo Fisher Scientific), PerCP/Cy5.5-conjugated anti-F4/80 antibody (BM8, BioLegend), Alexa Fluor 488-conjugated anti-CX$_3$CR1 antibody (R&D Systems, USA) and APC-conjugated anti-Ly6C antibody (AL-21 BD Biosciences). Intracellular staining of NOS2 and Arg1 was performed via surface staining with BV 510-conjugated anti-CD45 antibody (30-F11, BioLegend), BUV 737-conjugated CD11b antibody (M1/70 BD Biosciences), BUV395-conjugated anti-CD11c antibody (HL3, BD Biosciences) and APC-conjugated anti-F4/80 antibody (BM8, BioLegend). Cells were fixed and permeabilised using the Foxp3/Transcription Factor Staining Buffer Set (Thermo Fisher Scientific) and stained with PE-conjugated anti-NOS2 antibody (CXNFT Thermo Fisher Scientific) and FITC-conjugated anti-Arg1 antibody (R&D Systems). CD4$^+$Foxp3$^+$ cells were surface stained with BUV395-conjugated anti-CD45 antibody (30-F11, BD Biosciences), BUV737-conjugated, anti-CD3e antibody (145–2C11, BD Biosciences) and BV510-conjugated, anti-CD4(RM4–5, BioLegend). After surface staining, cells were fixed and permeabilised using the Foxp3/Transcription Factor Staining Buffer Set (Thermo Fisher Scientific) and stained with PerCP-Cy5.5-conjugated anti-Foxp3 antibody (FJK-16s, Thermo Fisher Scientific). Dead cells were detected using Fixable Viability Stain 780 (BD Biosciences). The cells were analysed using the LSR II flow cytometer (BD Biosciences). Gating strategies used for flow cytometry analysis were shown in Supplementary Fig. 8.

**BMDMs culturing**. Bone marrow cells were isolated from the femurs and tibias of the C57BL/6 J male mice. Cells were seeded in 12-well plates at $1 \times 10^6$ cells ml$^{-1}$ in complete-RPMI medium (RPMI-1640 medium supplemented with 10% foetal bovine serum, 10 mM HEPES, 20 μM GlutaMAX I (Fisher Scientific), 50 μM 2-mercaptoethanol (Wako), 100 U ml$^{-1}$ penicillin, 100 μg ml$^{-1}$ streptomycin) supplemented with 20 ng ml$^{-1}$ macrophage colony-stimulating factor (BioLegend). The medium in the wells was replaced with fresh medium on day 3. On day 6, BMDMs were harvested and analysed via flow cytometry as mentioned above. The putrescine group was treated with 100 μM putrescine from day 1 to day 6. For the putrescine tracing experiment, $^{15}$N$_2$-putrescine 2·HCl with atomic purity >98% (Sigma-Aldrich) was solubilised in saline and neutralised to pH 7.0. For the putrescine tracing experiment: On day 6, BMDMs were cultured with 100 μM stable isotope-labelled putrescine for 24 h. Then, BMDMs were washed and lysed with RIPA buffer, and isotope enrichment was analysed via GC-MS. For the evaluation of OXPHOS and hyp-eIF5A: From day 6 to day 9, BMDMs were cultured with macrophage colony-stimulating factor in the presence or absence of DFMO (1 mM) and putrescine (100 μM). Then, BMDMs were washed and lysed with RIPA buffer supplemented with protease inhibitor cocktail (Thermo Fisher Scientific). The lysate was use Western blotting analysis of hyp-eIF5A and OXPHOS.

**DSS-induced experimental colitis**. Eight-week-old mice treated with the SK930 or WT strains were treated with 2.0% DSS (MW 5000, Wako) aqueous solution sterilised through a 0.22-μm filter. To induce colitis, mice were treated with 2.0% DSS aqueous solution for 6 days followed by normal drinking water for 3 days, during which we observed survival rate and disease activity index (DAI)[68]. In a separate experiment, mice were sacrificed 5 days after treatment to evaluate the histology and macrophage population. Stool specimen rigidity scores were defined as follows: 0 = normal, 1 = soft but formed, 2 = soft, 3 = very soft, 4 = watery diarrhoea. The haematochezia scores were defined as follows: 0 = occult blood test-negative, 1 = occult blood test-positive, 2 = slight, 3 = blood traces in stool, 4 = gross rectal bleeding. Faecal occult blood was examined using ColoScreen-ES (Helena Laboratories, USA).

**Histological scoring**. Colonic tissue samples were obtained from 1 cm above the anus. The samples were embedded in paraffin and cut into 3-μm sections. The sections were deparaffinised, rehydrated, and stained with haematoxylin (Agilent Technologies) and eosin (Wako). A blinded observer scored the stained colonic tissue sections using a previously published system for the following measures[69]: crypt architecture (normal = 0; severe crypt distortion with loss of entire crypts = 3), degree of inflammatory cell infiltration (normal = 0; dense inflammatory infiltrate = 3), muscle thickening (base of crypt sits on the muscularis mucosae = 0; marked muscle thickening present = 3), goblet cell depletion (absent = 0; present = 1) and crypt abscess (absent = 0; present = 1). Three randomly selected areas were examined per slide. The histological damage score was the sum of each individual score.

**Measurement of lipocalin-2**. Frozen faeces (approximately 50 mg) were diluted 10-fold with 0.1% Tween-20/D-PBS and vortexed for 20 min. The upper aqueous portion was collected by centrifugation (12,000 × g for 10 min at 4 °C). Lipocalin-2 was measured in the faecal extraction using a Mouse Lipocalin-2/NGAL Quantikine ELISA Kit® (R&D Systems) following the manufacturer's protocol.

**Statistical analysis**. Prism software, ver. 8.0 (GraphPad Software Inc., USA) was used, otherwise specified, for statistical analysis. The differences between two or more groups were analysed using the two-tailed Student's t-test or ANOVA followed by Tukey's multiple comparison test. When the variances were not homogeneous, the data were analysed by Welch's t-test or Welch's ANOVA followed by

Dunnett's T3 method. When the data distribution was not normal, a Mann–Whitney U-test or Kruskal–Wallis test followed by Steel-Dwass test (R software). All correlation analyses were performed using Spearman's correlation test. The survival curve was analysed using a log-rank test. Statistically significant differences are indicated as $*P < 0.05$, $**P < 0.01$ and $***P < 0.001$. In the bar graphs, bars indicate the means, and error bars indicate the standard error of the mean. In all summaries, dot plots indicate data from individual samples.

**Reporting summary**. Further information on research design is available in the Nature Research Reporting Summary linked to this article.

## Data availability
The data supporting the findings of this study are available in the main article and supplementary files or from the corresponding author upon reasonable request. Source data are provided as a Source Data file.

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

## Acknowledgements

We are grateful to Prof. Patrick Woster for providing the ODC inhibitor DFMO. We thank Ayano Yamashita, Yusuke Kitada and Yumiko Fujimura for assistance with the animal experiments. We thank Tomohiro Ishimaru for the established conditions of immunostaining for autophagy. This study was supported by grants from the Japan Society for the Promotion of Science (17KT0055, 16H01369, 18H04680, 20H00509, and 20H05876 to KH, 18H04805 to S.F, 20H00575 to MM, and 20K19738 to AK), AMED-Crest (16gm1010004h0101, 17gm1010004h0102, 18gm1010004h0103, 19gm1010004s0104, 20gm1010004h0105, and 20gm1310009h0001 to KH, JP19gm1010009 to S.F.), AMED (18ek0109303h0001 to KH), JST PRESTO (JPMJPR1537 to S.F.), JST ERATO (JPMJER1902 to S.F.), Yakult Foundation (KH), Keio Gijuku Academic Development Funds (KH), the SECOM Science and Technology Foundation (KH), the Takeda Science Foundation (S.F., K.H.), the Food Science Institute Foundation (S.F.), the Program for the Advancement of Research in Core Projects under Keio University's Longevity Initiative (S.F.), The Cell Science Research Foundation (KH), The Science Research Promotion Fund, and The Promotion and Mutual Aid Corporation for Private Schools of Japan (KH).

## Author contributions

A.N. performed the gnotobiote experiments, analysis of polyamine concentrations, and in vitro experiments using BMDMs and intestinal organoids; S. Ku. Designed and prepared the genetically modified *E. coli* experiments; Y.O. performed the exGF mice

experiments; S.F. performed metabolome analysis; D.T., Y.N. and Y.F. supported flow cytometry and the in vitro experiments; S. Ki. performed histological examination; W.O. performed the intestinal organoids experiments, and OXPHOS western blotting. M.O. helped in vitro experiments using BMDMs. Y.S. and S.S. supported autophagy experiments. A.K. performed immunohistochemical analysis of autophagy; A.N., M.M. and K.H. designed the study and interpreted the data; A.N. and M.M. wrote the manuscript; K.H. critically revised the manuscript.

## Competing interests

The authors declare no competing interests.
