## [Peer Review File · Nature Communications]

REVIEWER COMMENTS

Reviewer #1 (Remarks to the Author):

This well-written and informative paper describes the effect of bacterial-derived polyamines on gut physiology and colitis. The authors have shown that FTM from conventional to GF mice increases the abundance of polyamines (both luminal and intracellular) and triggers CEC proliferation. This phenotype could be reproduced in F1 mice harbouring WT *E. coli*, but not *E. coli* SK930, and using exogenous polyamines in organoids. Mechanistically WT, but not SK930, significantly increased hypusination of eIF5A in CECs, thus activating the translation elongation factor, which was inhibited by N1-monoguanyl 1,7-diaminoheptane. Moreover, WT and SK930 differentially affected differentiation of M1/M2 macrophages and the former had a protective phenotype in the DSS model.

The experimental design and analysis are robust and the paper describes novel data, which supports the conclusions.

Minor comments:

1. The paper would benefit from an introduction section, encompassing polyamines, hypusination, eIF, etc.
2. It is not clear if the WT and SK930 *E. coli* are isogenic strains; two references (10 and 18) are given for SK930.
3. Where do WT and SK930 *E. coli* reside in the gut? Is it the cecum? This is important, as many of the measurements have been made in the colon. This requires some discussion.
4. Do WT and SK930 shed similarly from DSS-treated mice?

Reviewer #2 (Remarks to the Author):

What percentage of polyamines come from microbes and what from the diet, because if the dietary source is significantly greater it begs the question - so what?, why study the role of microbially derived polyamines, is this study just dressing up something to be important that is actually not. If the authors can make a better case for why we should look at this source I would be better convinced. Using *E. coli* as the source of polyamines has no bearing on the colon (even if they are there in early life, they are there for a very short time), and also very little relevance to the small intestine, where the majority of polyamines will come from diet. What other species make polyamines?

Abstract: findings suggest nothing – please change to we “conclude that”

Line 23-24 – why does “rapid epithelial turnover necessitate(s) microbial colonization in the gut”? why is it necessary to have microbial colonization of the gut as a consequence of rapid epithelial turnover?

Line 39-41 – what diet was being used in these mice – could this have led to the difference?

Line 41-44 – why would they not be? The host cannot differentiate between polyamines from diet and from a bacterium. This question is akin to asking if butyrate derived from commensal bacteria is incorporated into colonocytes – why would any bacterially-derived molecule that is in the diet not be? Is there anything chemically different between polyamines from the diet and from bacteria that would justify this work and hypothesis? In fact the authors make my point for me – lines 67-71: The authors do not use bacterially derived 15N polyamines for their organoid experiments, so the source of a the polyamine is irrelevant, so before they undertook this work, they had already

answered the question and proved the hypothesis – whether the polyamine is from Sigma-Aldrich, bacteria or diet, it doesn't matter – CECs will use it.

Line 52-53: If the SK930 mutant was not able to make putrescine, why was it not absent in the culture supernatant, where was the low level of this polyamine coming from? Where are the counts of CFU of the E. coli pre and post experiment – did the numbers vary in the gut of the animals?

Line 328: Culture of bacteria: While I'm here, what were the growth characteristics of the SK930 mutant in a minimal media, where they need to make much more from less – e.g. just glucose – since the animal model's gut is not full of LB this may influence the colonization of the mouse gut by the mutant. Also what is the fitness of SK930 versus the WT? Another key point that is not address is the total metabolome of the SK930 vs the WT E. coli – are there are other anti-inflammatory molecules, being produced by the WT vs SK930, and if the WT is at higher numbers in the gut, could these also be at higher concentrations too – how can you say this is all due to putrescine and polyamines - E coli can make acetate too and this can regulate inflammasome activity (see <https://www.nature.com/articles/s12276-019-0276-5>). The heatmap in Supp Fig 1C, if I read it correctly is for an FMT from a conventional SPF mouse which has a complex metabonome, into the GF mouse, but there are no metabolite heatmaps of the GF mouse colonized with WT vs SK930.

Line 82-83: I think it is not correct to differentiate between commensal and non-commensal derived polyamines – since there is no difference – in the text above (Lines 67-71) you've already proved it does not matter where it comes from as the host, in this case organoids, do not differentiate between different sources.

Line 142-144: While this statement may be true, I don't think anyone ever thought they weren't incorporated by host cells, in the same sense that any bacterially derived metabolite is, whether it is acetate, butyrate, BCFA or BCAA. While the work is nice, the authors have shown that is does not matter where the polyamine comes from, since they did not use bacterially derived polyamines for the organoid work – so the source is irrelevant.

Reviewer #3 (Remarks to the Author):

Hase and colleagues describe that bacterial derived putrescine is taken up by colon epithelial cells in mice to enhance barrier function and immunity in the gut. The paper is interesting and technically well done but lacks a couple of control experiments and additional mechanistic insights.

1. For example, although the authors show that eIF5A hypusination may be part of the mechanism, the corresponding downstream effects like mitochondrial respiration and autophagy induction remain unexplored. It should at least be tested, whether autophagy is enhanced in CECs upon decoration with putrescin producing E. Coli and ideally they would repeat some key experiments in ATG deficient mice. Also, OXPHOS proteins should be determined (and ideally respiration itself) under the same conditions.

2. In this line, some crucial in vivo experiments should be repeated with N1-monoguanyl 1,7-diaminoheptane, in order to show causality of hypusination for e.g. proliferation effects.

3. The bioavailability of polyamines is a crucial point for the whole manuscript. Therefore N-labelled putrescine should be administered also in vivo (not only in organoids) in order to demonstrate, that it transfers into the colon epithelial cells (and/or even in the system, like blood serum).

4. Survival rates in fig 4 C look interesting, but how does the curve proceed after 9 days?

Reviewer #4 (Remarks to the Author):

In the manuscript entitled "Symbiotic polyamine metabolism regulates mucosal barrier and immunity in the gut", Atsuo Nakamura and colleagues describe very interesting new insights into the role of bacteria-derived colonic putrescine in the maintenance of colonic tissue homeostasis. Taking advantage of a putrescine-deficient *E. coli* SK930 strain and SK930-strain-associated mice, the authors describe the effects of locally released putrescine on colonic epithelial cells and intestinal macrophages and define its overall impact on the development of DSS-induced colitis. Although earlier studies already indicated a potential involvement of polyamines in IBD, experimental colitis, intestinal immunity and macrophage differentiation, the here presented data significantly improve our understanding of the underlying mode of action and identify eIF5A as an important intracellular downstream target of putrescine in intestinal epithelial cells. To my knowledge, this is the first experimental proof, that intestinal epithelial cells are able to uptake exogenous putrescine and convert it to spermidine.

Overall, this comprehensively written, well structured and very interesting manuscript relevantly improves our understanding of the molecular interplay between the luminal microbiome, the gut epithelium and mucosal immune cells in the context of intestinal tissue homeostasis and/or inflammation.

To further improve the overall impact of the study and underline its novelty, I recommend addressing the following aspects:

- The title of the manuscript should point more specifically to the investigated cell types and to the identified mode of action. The title states that polyamines regulate mucosal barrier function, although the impact of putrescine on the epithelial barrier integrity has not been analyzed on a functional level (e.g. via FITC-Dextran Assay).
- Regarding Figure 1F, the authors state that the proliferation of epithelial cells was significantly enhanced in WT-strain-associated mice compared to the GF- and SK930-strain-associated mice, but they do not comment on the fact that the proliferation of epithelial cells in SK930-strain-associated mice was significantly higher than in GF mice. It would be good to mention that obviously there is also a putrescine-independent proliferation-inducing effect of the transferred *E. coli* strain.
- The term "hypusination" has to be introduced in more detail. Does there already exist any published data, which might suggest a potential role of eIF5A in IBD pathogenesis? This aspect should be included in the discussion.
- The representative Western blot image depicted in Figure 2f shows only a minor regulation of hyp-eIF5A in MCHA-exposed organoids, which is hard to recognize at all. Are the authors able to prove that this minor regulatory effect indeed functionally underlies the impaired growth of organoids?
- There is a discrepancy in Figure 2g between the depicted representative Western blot image and the summarizing graph: The image seems to indicate higher levels of hyp-eIF5A in epithelial cells of SK930-strain-associated mice than in epithelial cells of GF mice, which is not reflected by the summarizing graph.
- In Figure 2h, the authors demonstrate that the presence of GC7 (inhibitor of hyp-eIF5A synthesis) resulted in a decreased growth of organoids. It would be important to confirm the successful inhibition of hyp-eIF5A in this experimental setting. Are the authors able to exclude that the observed effect on organoid growth might be due to any unspecific toxic effect of the inhibitor?
- The authors hypothesize that it might be possible that exogenous polyamine may also regulate CX3CR1^{high} Ly6C⁻ macrophage differentiation through the metabolic reprogramming to OXPHOS by hypusinated eIF5A. This is very speculative. In order to meet the requirements of a high-ranking publication, this hypothesis needs to be validated by experimental analyses. Are the authors able to show altered hyp-eIF5A levels in BMDMs after exposure to putrescine? Are they able to detect an altered expression of OXPHOS-related proteins in BMDMs cultured in the presence/absence of putrescine? Are these potential effects dependent on the presence of hypusinated eIF5A? Are BMDMs able to uptake exogenous putrescine?

- The described observation that WT-strain-associated mice were more resistant to DSS-induced colitis than SK930-strain-associated mice is somehow in accordance with the following recently published study, which described that spermidine protects mice during DSS-induced colitis: Gobert AP, Al-Greene NT, Singh K, et al. Distinct Immunomodulatory Effects of Spermine Oxidase in Colitis Induced by Epithelial Injury or Infection. *Front Immunol.* 2018;9:1242. Thus, it is very important to mention and discuss this study. Moreover, another recent study indicated that polyamines promote the differentiation of regulatory T cells (Carriche GM, Almeida L, Stüve P, et al. Regulating T cell differentiation through the polyamine spermidine. *J Allergy Clin Immunol.* 2020). Are the authors able to confirm experimentally that the identified effects of polyamines on intestinal epithelial cells and/or macrophages indeed underlie the observed differences in the severity of colitis between WT-strain-associated mice and SK930-strain-associated mice?

Minor concerns:

- The assignment of b and c in the legend of Supplementary Figure 3 is not correct.
- In Figure 3d, an isotype control staining should be provided in order to confirm the correct gating of the NOS2-expressing cell population.

NCOMMS-20-10409-T

"Symbiotic polyamine metabolism regulates epithelial proliferation and macrophage differentiation in the gut."

We sincerely appreciate the reviewers' valuable comments, which helped us greatly improve our manuscript. Below are the point-by-point responses to the reviewers' concerns and the direction of the revision. We thank the reviewers for their helpful suggestions, which led to a much improved manuscript. To address the critical points raised, we performed several additional experiments. For each point, the reviewer's comments are in ***bold italic***, and our responses are in regular text. We reformed our manuscript from a letter style to a full paper composed of the introduction, results and discussion, and conclusion sections. All modifications made to the manuscript are underlined to indicate their location.

Reviewer #1 (Remarks to the Author):

This well-written and informative paper describes the effect of bacterial-derived polyamines on gut physiology and colitis. The authors have shown that FTM from conventional to GF mice increases the abundance of polyamines (both luminal and intracellular) and triggers CEC proliferation. This phenotype could be reproduced in F1 mice harbouring WT E. coli, but not E. coli SK930, and using exogenous polyamines in organoids. Mechanistically WT, but not SK930, significantly increased hypusination of eIF5A in CECs, thus activating the translation elongation factor, which was inhibited by N1-monoguanyl 1,7-diaminoheptane. Moreover, WT and SK930 differentially affected differentiation of M1/M2 macrophages and the former had a protective phenotype in the DSS model.

The experimental design and analysis are robust and the paper describes novel data, which supports the conclusions.

Thank you for your constructive comments and positive evaluation of our paper. The comments have all been addressed and are described in detail below.

Minor comments:

1. The paper would benefit from an introduction section, encompassing polyamines, hypusination, eIF, etc.

We further described polyamines and hypusination of eIF5A in the introduction of the revised manuscript (lines 19–37) as follows: “Polyamines, which are aliphatic compounds containing more than two amino groups, are widely distributed among prokaryotes and eukaryotes, including humans, and regulate multiple biological processes, including translation, transcription and cell proliferation and differentiation. Among polyamines, putrescine and spermidine are especially prevalent in the large intestine of human and mice. We previously reported that the abundance of luminal polyamines is positively correlated with longevity in mice. Furthermore, the enhancement of luminal putrescine production by administration of probiotics and arginine improved reactive hyperemia index representing endothelial function in randomized controlled clinical trial. These observations illustrate that exogenous polyamines may possess a substantial impact on the host physiology.

In mammalian cells, putrescine is biosynthesized by ornithine decarboxylase (ODC) from precursor ornithine, then spermidine is generated from putrescine by addition of aminopropyl groups derived from decarboxylated S-adenosyl methionine. Spermidine serves as a substrate for

hypusine, in which the aminobutyl group of spermidine is attached to a specific lysine residue. Hypusination of eukaryotic translation initiation factor 5A (eIF5A) is required for its activation, and hypusinated eIF5A (hyp-eIF5A) is involved in translation initiation, elongation, termination, and cell cycles.”

2. It is not clear if the WT and SK930 *E. coli* are isogenic strains; two references (10 and 18) are given for SK930.

Reference #25 (Kitada et al. *Sci. Adv.* 4:eaat0062, 2018) is the first report of *E. coli* SK930. Therefore, we cited only this article when mentioning the SK930 strain in the revised manuscript.

3. Where do WT and SK930 *E. coli* reside in the gut? Is it the caecum? This is important, as many of the measurements have been made in the colon. This require some discussion.

Thank you for these questions and comments. We measured the bacterial CFUs in the luminal contents throughout the intestines of WT-strain-associated and SK930-strain-associated mice. Although both strains were distributed widely from the jejunum to the distal colon, the bacterial density was higher in the large intestines (caecum and colon) than in the small intestines (jejunum and ileum). Importantly, the abundances of the two strains did not significantly differ in any of the regions tested here. These data are shown in Supplementary Fig. 3a:

4. Do WT and SK930 shed similarly from DSS-treated mice?

We quantified the bacterial numbers from the colonic contents of the DSS-treated gnotobiotic mice via quantitative PCR using *E. coli*-specific primers. The numbers of *E. coli* (cells/g wet faeces) were comparable between the WT-strain-associated and SK930-strain-associated mice (Supplementary Fig. 7a). Furthermore, bacterial numbers in the colons remained unchanged before and after colitis development (Supplementary Figs. 3a and 7a). Updated Supplementary Fig. 7a:

Reviewer #2

1. *What percentage of polyamines come from microbes and what from the diet, because if the dietary source is significantly greater it begs the question - so what?, why study the role of microbially derived polyamines, is this study just dressing up something to be important that is actually not. If the authors can make a better case for why we should look at this source I would be better convinced. Using E. coli as the source of polyamines has no bearing on the colon (even if they are there in early life, they are there for a very short time), and also very little relevance to the small intestine, where the majority of polyamines will come from diet. What other species make polyamines?*

Thank you for raising this critical issue regarding the biological significance of microbial polyamines. To address these concerns, we examined the contribution of exogenous polyamines from the diet and from the microbiota to colonic physiology via the following experiments:

Experiment I: Delivery of dietary polyamines to the colonic lumen

We initially analysed whether dietary putrescine reaches the colon. SPF mice were orally gavaged with stable-isotope-labelled $^{15}\text{N}_2$ -putrescine in PBS containing a blue food colouring agent as a marker to determine the position of $^{15}\text{N}_2$ -putrescine in the digestive tract. We collected the blue-stained luminal contents in the lower small intestine (2 hours after oral administration; n = 5) and in the colon (3.5 hours after oral administration; n = 5) and quantified the isotope-labelled and total putrescine amounts via GC-MS. The isotope-labelled/total putrescine ratio was 35.0% in the small intestinal luminal contents 2 hours after oral administration. This ratio was steeply decreased in the colonic lumen, where the isotope-labelled putrescine represented only 0.36% of the total amount at 3.5 hours (Supplementary Fig. 4c in the revised manuscript). Thus, a large portion of dietary putrescine was absorbed or degraded in the upper digestive tract before reaching the colonic lumen. These data are included in Supplementary Fig. 4a–c:

Experiment II: Uptake of polyamines from the lumen into the colonic epithelium

To assess the uptake and use of luminal putrescine in the colonic epithelial cells (CECs), SPF mice were anaesthetised and injected with isotope-labelled $^{15}\text{N}_2$ -putrescine at the same amount as in Experiment I, into the colonic lumen (luminal administration model; Supplementary Fig. 4d). Two hours later, we collected the CECs to determine the uptake of isotope-labelled putrescine. We observed that 11.2% of the total putrescine was derived from the luminal putrescine in the luminal administration model (Supplementary Fig. 4e). We also measured the ratio of isotope-labelled putrescine in the CECs after oral administration (the same protocol as in Experiment I). The uptake of isotope-labelled putrescine at 3.5 hours after oral administration was much lower (2.23%) than that in the luminal administration model. Of note, a similar level of isotope-labelled putrescine (2.96%) was detected as early as 2 hours after oral administration, suggesting that some of the putrescine absorbed in the upper digestive tract may be translocated to the CECs via the bloodstream. Furthermore, isotope-labelled spermidine was more abundant in the CECs after luminal administration than after oral administration. Therefore, the colonocytes most likely used

the exogenous polyamines generated by the commensal bacteria in the colonic lumen rather than those obtained from the diet. We described these results in lines 123–154 in the revised manuscript. Updated Supplementary Fig. 4d–f:

2. What other species make polyamines?

Polyamines (putrescine, spermidine, and spermine) are ubiquitous in all organisms. We mentioned this in the text (lines 19–22) as follows: “Polyamines, which are aliphatic compounds containing more than two amino groups, are widely distributed among prokaryotes and eukaryotes, including humans, and regulate multiple biological processes, including translation, transcription and cell proliferation and differentiation”.

3. Abstract: findings suggest nothing – please change to we “conclude that”

We changed this in the revised manuscript (line 14).

4. Line 23-24 – why does “rapid epithelial turnover necessitate(s) microbial colonization in the gut”? why is it necessary to have microbial colonization of the gut as a consequence of rapid epithelial turnover?

We agree that the original sentence was incorrect; we amended it (lines 39–40) as follows: “Microbial colonisation induces rapid epithelial turnover in the gut.”

5. Line 39-41 – what diet was being used in these mice – could this have led to the difference?

In the faecal transplantation experiment, mice were fed sterilised (for GF mice) or non-sterilised (for exGF mice) water and roughage CE-2 pellet chow (Clea Japan, Inc.). We added this information to the Methods (lines 511–513) of the revised manuscript: “Mice were provided with sterilised (for GF) or non-sterilised (for exGF and SPF) water and sterilised (for GF, exGF and SPF) CE-2 pellet chow (Clea Japan, Inc.)”.

6. Line 41-44 – why would they not be? The host cannot differentiate between polyamines from diet and from a bacterium. This question is akin to asking if butyrate derived from commensal bacteria is incorporated into colonocytes – why would any bacterially-derived molecule that is in the diet not be? Is there anything chemically different between polyamines from the diet and

from bacteria that would justify this work and hypothesis? In fact the authors make my point for me – lines 67-71: The authors do not use bacterially derived 15N polyamines for their organoid experiments, so the source of a the polyamine is irrelevant, so before they undertook this work, they had already answered the question and proved the hypothesis – whether the polyamine is from Sigma-Aldrich, bacteria or diet, it doesn't matter – CECs will use it.

We agree that the host does not distinguish between bacterial and dietary putrescine and that all putrescines are identical chemicals regardless of whether they are derived from the diet, intestinal bacteria, or Sigma-Aldrich. Previous studies showed that a large portion of dietary polyamines including putrescine is absorbed in the small intestine (Uda et al. *J Gastroenterol Hepatol* 18, 554–559, 2003, Milovic et al. *Eur J Gastroenterol Hepatol* 13, 1021-1025, 2001). We also confirmed that most orally delivered putrescine was absorbed in the small intestines (Supplementary Fig. 4a–c), whereas CECs mainly took up putrescine from the colonic lumen through absorption from the bloodstream at a lesser amount (Supplementary Fig. 4d–f). We previously reported that putrescine in the colonic lumen is derived from commensal gut bacteria (Matsumoto et al. *Sci Rep.* 4: 4548, 2014). Thus, we believe that bacterial putrescine is the major source of exogenous putrescine in CECs.

As indicated, we did not use bacterially derived putrescine in the intestinal organoid experiments. We used the intestinal organoids to examine whether CECs actively take up exogenous putrescine and metabolise it to spermidine.

7. Line 52-53: If the SK930 mutant was not able to make putrescine, why was it not absent in the culture supernatant, where was the low level of this polyamine coming from? Where are the counts of CFU of the E. coli pre and post experiment – did the numbers vary in the gut of the animals?

To determine the source of putrescine in the SK930 culture supernatant, we measured the polyamine concentration in the LB medium used for the culture. The concentrations of putrescine and spermidine in the LB medium were 6.13 μM and 19.8 μM , respectively. These values are equivalent to those in the SK930-strain culture supernatant (2.15 and 19.8 μM) as shown in Supplementary Fig. 2a:

Thus, the low levels of polyamines detected in the culture supernatant of the SK930 strain were most likely derived from the LB medium. We mentioned this in lines 101-104 in the revised manuscript as follows: “Putrescine was nearly absent in the SK930 culture supernatant, and spermidine concentrations in the wild-type (WT) and SK930 culture supernatants were comparable to that in the Luria-Bertani (LB) medium (Supplementary Fig. 2a).”

In the bacterial culture, 1×10^2 CFU/ml of each strain was added to the medium. After culturing for 24 hours, both bacterial numbers reached 1×10^6 CFU/ml. This information is included in the legend in Supplementary Fig. 2b.

We also counted bacterial CFUs in various regions of the intestines and confirmed that none

of the regions differed between the two gnotobiotic groups associated with the WT or SK930 strains. These data are included in Supplementary Fig. 3a:

8. Line 328: Culture of bacteria: While I'm here, what were the growth characteristics of the SK930 mutant in a minimal media, where they need to make much more from less – e.g. just glucose – since the animal model's gut is not full of LB this may influence the colonization of the mouse gut by the mutant. Also, what is the fitness of SK930 versus the WT? Another key point that is not address is the total metabolome of the SK930 vs the WT *E. coli* – are there are other anti-inflammatory molecules, being produced by the WT vs SK930, and if the WT is at higher numbers in the gut, could these also be at higher concentrations too – how can you say this is all due to putrescine and polyamines - *E coli* can make acetate too and this can regulate inflammasome activity (see <https://www.nature.com/articles/s12276-019-0276-5>). The heatmap in Supp Fig 1C, if I read it correctly is for an FMT from a conventional SPF mouse which has a complex metabonome, into the GF mouse, but there are no metabolite heatmaps of the GF mouse colonized with WT vs SK930.

We understand this concern. To confirm the fitness of the SK930 strain *in vivo*, we examined the CFUs throughout the intestines. As shown above (Supplementary Fig. 3a), the abundances of the two strains did not significantly differ in any of the regions tested here. Additionally, GC-MS analysis showed that the faecal concentrations of short-chain fatty acids (i.e., acetate, propionate, and butyrate) were similar between the gnotobiotic mice associated with both the WT and SK930 strains. These data are included in Supplementary Fig. 3c:

Furthermore, we performed CE-TOFMS-based target metabolomic profiling of the caecal contents ($n = 3/\text{group}$) as described previously (Yamashita et al. *PLoS One*. 9: e86426, 2014). We detected 38 metabolites in both groups, of which, only putrescine was markedly higher (20.4-fold) in the WT-associated mice than in the SK930-associated mice. Little to no differences were seen in the other luminal metabolites between the two groups. Thus, the SK930 strain retained a similar ability to that of the parental WT strain in growth and production of metabolites other than putrescine. The metabolomic data are included in Supplementary Table 3:

Metabolite	Concentration (µM)										WT / SK930	WT vs. SK930
	WT			SK			WT		SK			
	WT-1	WT-2	WT-3	SK-1	SK-2	SK-3	Mean	S.D.	Mean	S.D.		
Putrescine	139.87	175.27	170.82	9.83	6.05	7.89	161.99	19.28	7.92	1.89	20.44	0.13
Creatinine	514.84	411.69	596.75	219.06	120.70	150.37	507.76	92.74	163.38	50.45	3.11	0.13
S-Adenosylmethionine	18.44	19.98	20.31	11.61	8.17	12.42	19.58	0.99	10.73	2.25	1.82	0.13
Gly	1461.85	1010.23	1127.21	750.90	583.98	683.40	1199.76	234.39	672.76	83.97	1.78	0.37
3-Hydroxybutyric acid	143.28	328.26	259.19	98.80	54.43	N.D.	243.58	93.47	76.62	31.37	3.18	0.37
His	369.79	256.04	232.97	175.91	134.40	161.37	286.27	73.24	157.22	21.06	1.82	0.37
Lactic acid	1238.05	884.21	704.82	543.35	440.58	401.97	942.36	271.33	461.97	73.07	2.04	0.37
Choline	1572.20	2762.94	1900.30	1057.95	841.40	841.40	2078.48	615.04	913.58	125.02	2.28	0.37
Asn	70.66	42.16	37.17	32.06	18.16	17.04	50.00	18.07	22.42	8.37	2.23	0.37
Gluconic acid	68.96	106.96	188.51	222.65	240.47	168.19	121.48	61.08	210.44	37.65	0.58	0.37
Citrulline	17.66	9.46	13.57	8.99	6.55	7.43	13.56	4.10	7.66	1.24	1.77	0.37
Ser	237.56	155.53	126.56	105.34	85.24	85.32	173.22	57.58	91.97	11.58	1.88	0.37
Val	1409.66	953.79	763.81	733.02	499.01	546.99	1042.42	331.92	593.00	123.60	1.76	0.37
Ornithine	27.89	11.23	27.78	10.89	7.33	15.19	22.30	9.59	11.14	3.93	2.00	0.37
Thr	126.06	50.71	51.13	43.30	21.26	24.07	75.97	43.38	29.54	12.00	2.57	0.37
Betaine aldehyde_+H ₂ O	20.39	43.62	34.49	22.23	17.27	21.27	32.84	11.70	20.26	2.63	1.62	0.37
Ala	1805.12	895.29	739.60	736.46	410.71	474.24	1146.67	575.52	540.47	172.67	2.12	0.37
Ile	980.47	584.01	448.08	498.10	324.13	327.92	670.85	276.62	383.38	99.37	1.75	0.37
Tyr	640.93	290.98	258.75	231.52	136.45	167.77	396.89	211.96	178.58	48.45	2.22	0.37
Leu	1880.37	819.40	648.25	653.96	325.63	337.88	1116.01	667.47	439.16	186.13	2.54	0.37
Creatine	692.35	553.77	964.38	614.92	489.14	483.05	736.83	208.89	529.04	74.44	1.39	0.37
Phe	834.97	345.02	297.17	288.14	145.38	159.17	492.38	297.65	197.56	78.75	2.49	0.37
Citric acid	174.77	141.70	167.35	147.42	142.66	142.88	161.27	17.35	144.32	2.68	1.12	0.37
Lys	1464.27	654.17	735.99	648.26	426.57	523.43	951.47	445.97	532.75	111.13	1.79	0.37
Asp	114.11	38.70	53.01	48.98	23.51	29.25	68.61	40.05	33.91	13.36	2.02	0.37
Met	543.43	197.25	154.40	173.24	88.02	93.51	298.36	213.31	118.25	47.70	2.52	0.37
Arg	1573.97	509.46	806.70	612.58	372.21	515.20	963.37	549.28	500.00	120.90	1.93	0.37
Gln	675.92	169.54	170.76	184.07	29.31	67.30	338.74	292.00	93.56	80.65	3.62	0.37
Pro	1783.05	1601.69	2713.40	1443.37	1232.06	1981.78	2032.71	596.43	1552.41	386.57	1.31	0.39
Trp	37.47	9.51	9.16	10.50	4.04	5.46	18.71	16.24	6.67	3.40	2.81	0.39
Glu	2853.49	1615.94	2037.91	2082.92	1113.68	2005.29	2169.11	629.12	1733.96	538.58	1.25	0.47
Succinic acid	1360.37	1507.97	345.51	1107.22	132.18	736.74	1071.28	632.86	658.71	492.18	1.63	0.47
Cys	25.36	9.80	6.95	10.92	9.16	7.14	14.04	9.91	9.07	1.89	1.55	0.51
Uridine	39.14	42.43	49.23	41.42	43.63	54.00	43.60	5.15	46.35	6.71	0.94	0.62
GABA	11.93	9.19	20.95	13.67	10.97	15.14	14.02	6.15	13.26	2.11	1.06	0.86
Hydroxyproline	9.02	6.54	15.98	7.62	N.D.	6.06	10.51	4.89	-	-	-	-
Spermine	11.77	14.52	16.27	14.58	N.D.	17.35	14.19	2.27	-	-	-	-
Spermidine	55.79	45.38	47.63	8.22	N.D.	N.D.	49.60	5.48	-	-	-	-

N.D.: not detected.

8. Line 82-83: I think it is not correct to differentiate between commensal and non-commensal derived polyamines – since there is no difference – in the text above (Lines 67-71) you've already proved it does not matter where it comes from as the host, in this case organoids, do not differentiate between different sources.

Following this suggestion, we amended the sentence (lines 171-172) as follows: “We subsequently explored the underlying mechanism by which polyamines regulate host epithelial proliferation.”

9. Line 142-144: While this statement may be true, I don't think anyone ever thought they weren't incorporated by host cells, in the same sense that any bacterially derived metabolite is, whether it is acetate, butyrate, BCFA or BCAA. While the work is nice, the authors have shown that it does not matter where the polyamine comes from, since they did not use bacterially derived polyamines for the organoid work – so the source is irrelevant.

As shown in Supplemental Fig. 4d–f, the colonocytes used putrescine from the colonic lumen. We agree that host cells can use exogenous polyamines regardless of their source. Based on our gnotobiotic studies (Fig. 1c–f) as well as the isotope-labelled putrescine challenge experiments (Supplementary Fig. 4), we think that dietary and bacterial polyamines respectively serve as the major exogenous sources in the small intestine and colon. Therefore, we modified the corresponding sentence (lines 310-312) as follows: “To our knowledge, this study provides the first direct evidence that host cells incorporate polyamines from the colonic lumen, which

eventually increase intracellular polyamine levels to regulate the mucosal epithelium and immunity.”

Reviewer #3 (Remarks to the Author):

Hase and colleagues describe that bacterial derived putrescine is taken up by colon epithelial cells in mice to enhance barrier function and immunity inn the gut. The paper is interesting and technically well done but lacks a couple of control experiments and additional mechanistic insights.

1. For example, although the authors show that eIF5A hypusination may be part of the mechanism, the corresponding downstream effects like mitochondrial respiration and autophay induction remain unexplored. It should at least be tested, whether autpaghy is enhanced in CECs upon decoration with putrescin producing E. Coli and ideally they would repeat some key experiments in ATG deficient mice. Also, OXPHOS proteins should be determined (and ideally respiration itself) under the same conditions.

Thank you for the positive evaluation of our paper and for raising these critical issues. Following the suggestion, we explored the impact of bacterial polyamines on autophagy and OXPHOS.

Autophagy: We could not conduct experiments with ATG-deficient mice because it takes considerable time and effort to introduce the mice and establish colonies in germ-free isolators. Alternatively, we evaluated autophagy activity in the CECs of gnotobiotic mice associated with WT or SK930 strains via immunofluorescent staining using antibodies for LC3 and SQSTM1/p62 (p62). We counted the positive dots in the CECs up to 50 μm from the bottom of the crypt, where the Ki67-positive cells were located (Fig. A).

Fig. A. Confocal microscopic image of immunostaining of the colon from a fasted mouse. Blue, green and red indicate the nucleus, LC3 and p62, respectively. Arrowheads indicate colocalisation of LC3 and p62.

The LC3-positive, p62-positive, and LC3/p62-double-positive signals represent autophagosomal formation, sequestosomal formation (which is selectively degraded by autophagy), and autophagosome/autolysosome formation (incorporation of p62-positive sequestosomes into autophagosomes), respectively. The WT-strain-associated mice exhibited significantly more p62-single- and LC3/p62 double-positive dots than did the SK930-associated mice. Further, the number of LC3-single-positive dots also tended to be increased in the WT-strain-associated mice. These data demonstrate that bacterial polyamines enhanced autophagy activity in the CECs. We included these data in the results section (lines 202–215) and Fig. 3a, b:

OXPHOS: We analysed expression levels of oxidative phosphorylation (OXPHOS) protein complexes I–V (CI–CV) in the CECs of gnotobiotic mice via western blotting. A previous study demonstrated that Hyp-eIF5A upregulated the mitochondrial OXPHOS protein complexes in macrophages (Puleston et al. *Cell Metab.* 30: 352–363, 2019). Conversely, CI–CV protein levels were lower in the *E. coli* from WT-associated (bacterial putrescine-sufficient) mice than in SK930-associated (bacterial putrescine-deficient) mice (Fig. 3c), indicating that CECs in SK930-associated mice depended more on OXPHOS-dependent energy metabolism than did those in WT-associated mice. This may reflect the different proliferative statuses of CECs between the two groups. Early works demonstrated that proliferating cell at the crypts depend more on glycolysis than on OXPHOS for ATP production, whereas the dependence of enterocytes on OXPHOS for their energy requirements increases as they differentiate and mature (Donohoe et al. *Mol Cell.* 48: 612–626, 2012; Rath et al. *Nat Rev Gastroenterol Hepatol.* 15: 497–516, 2018). We showed that colonization by the WT strain markedly increased the proliferating cells in the colonic crypts (Fig. 1f), which may eventually cause a Warburg-like effect, overwhelming the hyp-eIF5a-OXPHOS axis. However, further investigation is required to clarify this. These data are included in the results section (lines 216–228) and Fig. 3c:

2. In this line, some crucial *in vivo* experiments should be repeated with N1-monoguanyl 1,7-diaminoheptane, in order to show causality of hypusination for for e.g. proliferation effects.

Following the suggestion, we administered 500 μ M N1-monoguanyl 1,7-diaminoheptane (GC7) to SPF mice in their drinking water. Hyp-eIF5A levels in the CECs were significantly reduced in the GC7-treated group, whereas those in the untreated control group remained unchanged. Correspondingly, GC7 treatment significantly reduced the number of EdU-positive proliferating cells. Thus, inhibiting hypusination suppressed the CEC proliferation both *in vivo* and *in vitro*, supporting the idea that polyamine-dependent eIF5A hypusination is essential to accelerate CEC proliferation. These data are included in the results section (lines 197–201) and Fig. 2j, k:

3. The bioavailability of polyamines is a crucial point for the whole manuscript. Therefore N-labelled putrescine should be administered also *in vivo* (not only in organoids) in order to demonstrate, that it transfers into the colon epithelial cells (and/or even in the system, like blood serum).

Thank you for raising a critical issue on the bioavailability of polyamines. To assess the uptake and usage of lumenally derived exogenous putrescine into CECs, SPF mice were anaesthetised and injected with isotope-labelled $^{15}\text{N}_2$ -putrescine into the colonic lumen (Supplementary Fig. 4d). Two hours later, we collected the CECs to determine the uptake of isotope-labelled putrescine. We observed that 11.2% of the total putrescine was derived from luminal putrescine (Supplementary Fig. 4e). We also measured the ratio of isotope-labelled putrescine in the CECs after oral administration. The uptake of isotope-labelled putrescine at 3.5 hours after oral administration was much lower (2.23%) than that in the luminal-administration model. Furthermore, isotope-labelled spermidine in the CECs was more abundant after luminal administration than after oral administration. These results imply that colonocytes use exogenous polyamines that are mainly generated by commensal bacteria in the colonic lumen rather than polyamines obtained from the diet. These data are included in Supplementary Figs. 4d and 4e:

We also analysed isotope-labelled putrescine in the portal blood in the luminal-administration model (Supplementary Fig. 4d). Approximately 40% of the total putrescine in the portal vein was derived from the colonic lumen. Thus, luminal putrescine was translocated into the portal vein. These data are included in Supplementary Fig. 4f:

4. Survival rates in fig 4 C look interesting, but how does the curve proceed after 9 days?

Although it would have been interesting to observe the survival curve after 9 days, we could not perform additional experiments on the survival rates of mice with DSS-induced colitis owing to the limited availability of the gnotobiotic mice. We do not believe that the lack of this experiment compromised our conclusions.

Reviewer #4 (Remarks to the Author):

In the manuscript entitled “Symbiotic polyamine metabolism regulates mucosal barrier and immunity in the gut”, Atsuo Nakamura and colleagues describe very interesting new insights into the role of bacteria-derived colonic putrescine in the maintenance of colonic tissue homeostasis. Taking advantage of a putrescine-deficient E. coli SK930 strain and SK930-strain-associated mice, the authors describe the effects of locally released putrescine on colonic epithelial cells and intestinal macrophages and define its overall impact on the development of DSS-induced colitis. Although earlier studies already indicated a potential involvement of polyamines in IBD, experimental colitis, intestinal immunity and macrophage differentiation, the here presented data significantly improve our understanding of the underlying mode of action and identify eIF5A as an important intracellular downstream target of putrescine in intestinal epithelial cells. To my knowledge, this is the first experimental proof, that intestinal epithelial cells are able to uptake exogenous putrescine and convert it to spermidine.

Overall, this comprehensively written, well structured and very interesting manuscript relevantly improves our understanding of the molecular interplay between the luminal microbiome, the gut epithelium and mucosal immune cells in the context of intestinal tissue homeostasis and/or inflammation.

To further improve the overall impact of the study and underline its novelty, I recommend addressing the following aspects:

Thank you for the positive evaluation of our paper and the constructive comments for improving our manuscript.

1. The title of the manuscript should point more specifically to the investigated cell types and to the identified mode of action. The title states that polyamines regulate mucosal barrier function, although the impact of putrescine on the epithelial barrier integrity has not been analyzed on a functional level (e.g. via FITC-Dextran Assay).

Because we did not directly verify the mucosal barrier function, we changed the title as follows: “Symbiotic polyamine metabolism regulates epithelial proliferation and macrophage differentiation in the gut.”

2. Regarding Figure 1F, the authors state that the proliferation of epithelial cells was significantly enhanced in WT-strain-associated mice compared to the GF- and SK930-strain-associated mice, but they do not comment on the fact that the proliferation of epithelial cells in SK930-strain-associated mice was significantly higher than in GF mice. It would be good to mention that obviously there is also a putrescine-independent proliferation-inducing effect of the transferred E. coli strain.

Thank you for this valuable suggestion. *E. coli* likely produces various metabolites other than polyamines. Our metabolomic analysis detected 38 metabolites in the caecal contents of the WT- and SK930-associated gnotobiotic mice (Supplementary Table 3). Among these *E. coli*-derived metabolites, lactate has been shown to stimulate CEC proliferation and maintain intestinal stemness (Okada et al., *Nat. Commun.* 4, 1654, 2013; Lee et al., *Cell Host Microbe.* 24, 833-846, 2018). Additionally, the LPS in the *E. coli* may induce epithelial cell proliferation (Olaya et al., *In Vitro Cell Dev Biol Anim.* 35: 43-48, 1999). We mentioned these possibilities in the results section (lines 117–120) as follows: “SK930 colonisation also slightly increased the CEC proliferation likely due to the stimulation from LPS and/or lactate (Supplementary Table 3), both

of which have been reported to accelerate CEC proliferation.”

Updated Supplementary Table 3:

Metabolite	Concentration (μM)										WT / SK930	WT vs. SK930
	WT			SK			WT		SK			
	WT-1	WT-2	WT-3	SK-1	SK-2	SK-3	Mean	S.D.	Mean	S.D.		
Putrescine	139.87	175.27	170.82	9.83	6.05	7.89	161.99	19.28	7.92	1.89	20.44	0.13
Creatinine	514.84	411.69	596.75	219.06	120.70	150.37	507.76	92.74	163.38	50.45	3.11	0.13
S-Adenosylmethionine	18.44	19.98	20.31	11.61	8.17	12.42	19.58	0.99	10.73	2.25	1.82	0.13
Gly	1461.85	1010.23	1127.21	750.90	583.98	683.40	1199.76	234.39	672.76	83.97	1.78	0.37
3-Hydroxybutyric acid	143.28	328.26	259.19	98.80	54.43	N.D.	243.58	93.47	76.62	31.37	3.18	0.37
His	369.79	256.04	232.97	175.91	134.40	161.37	286.27	73.24	157.22	21.06	1.82	0.37
Lactic acid	1238.05	884.21	704.82	543.35	440.58	401.97	942.36	271.33	461.97	73.07	2.04	0.37
Choline	1572.20	2762.94	1900.30	1057.95	841.40	841.40	2078.48	615.04	913.58	125.02	2.28	0.37
Asn	70.66	42.16	37.17	32.06	18.16	17.04	50.00	18.07	22.42	8.37	2.23	0.37
Gluconic acid	68.96	106.96	188.51	222.65	240.47	168.19	121.48	61.08	210.44	37.65	0.58	0.37
Citrulline	17.66	9.46	13.57	8.99	6.55	7.43	13.56	4.10	7.66	1.24	1.77	0.37
Ser	237.56	155.53	126.56	105.34	85.24	85.32	173.22	57.58	91.97	11.58	1.88	0.37
Val	1409.66	953.79	763.81	733.02	499.01	546.99	1042.42	331.92	593.00	123.60	1.76	0.37
Ornithine	27.89	11.23	27.78	10.89	7.33	15.19	22.30	9.59	11.14	3.93	2.00	0.37
Thr	126.06	50.71	51.13	43.30	21.26	24.07	75.97	43.38	29.54	12.00	2.57	0.37
Betaine aldehyde +H ₂ O	20.39	43.62	34.49	22.23	17.27	21.27	32.84	11.70	20.26	2.63	1.62	0.37
Ala	1805.12	895.29	739.60	736.46	410.71	474.24	1146.67	575.52	540.47	172.67	2.12	0.37
Ile	980.47	584.01	448.08	498.10	324.13	327.92	670.85	276.62	383.38	99.37	1.75	0.37
Tyr	640.93	290.98	258.75	231.52	136.45	167.77	396.89	211.96	178.58	48.45	2.22	0.37
Leu	1880.37	819.40	648.25	653.96	325.63	337.88	1116.01	667.47	439.16	186.13	2.54	0.37
Creatine	692.35	553.77	964.38	614.92	489.14	483.05	736.83	208.89	529.04	74.44	1.39	0.37
Phe	834.97	345.02	297.17	288.14	145.38	159.17	492.38	297.65	197.56	78.75	2.49	0.37
Citric acid	174.77	141.70	167.35	147.42	142.66	142.88	161.27	17.35	144.32	2.68	1.12	0.37
Lys	1464.27	654.17	735.99	648.26	426.57	523.43	951.47	445.97	532.75	111.13	1.79	0.37
Asp	114.11	38.70	53.01	48.98	23.51	29.25	68.61	40.05	33.91	13.36	2.02	0.37
Met	543.43	197.25	154.40	173.24	88.02	93.51	298.36	213.31	118.25	47.70	2.52	0.37
Arg	1573.97	509.46	806.70	612.58	372.21	515.20	963.37	549.28	500.00	120.90	1.93	0.37
Gln	675.92	169.54	170.76	184.07	29.31	67.30	338.74	292.00	93.56	80.65	3.62	0.37
Pro	1783.05	1601.69	2713.40	1443.37	1232.06	1981.78	2032.71	596.43	1552.41	386.57	1.31	0.39
Trp	37.47	9.51	9.16	10.50	4.04	5.46	18.71	16.24	6.67	3.40	2.81	0.39
Glu	2853.49	1615.94	2037.91	2082.92	1113.68	2005.29	2169.11	629.12	1733.96	538.58	1.25	0.47
Succinic acid	1360.37	1507.97	345.51	1107.22	132.18	736.74	1071.28	632.86	658.71	492.18	1.63	0.47
Cys	25.36	9.80	6.95	10.92	9.16	7.14	14.04	9.91	9.07	1.89	1.55	0.51
Uridine	39.14	42.43	49.23	41.42	43.63	54.00	43.60	5.15	46.35	6.71	0.94	0.62
GABA	11.93	9.19	20.95	13.67	10.97	15.14	14.02	6.15	13.26	2.11	1.06	0.86
Hydroxyproline	9.02	6.54	15.98	7.62	N.D.	6.06	10.51	4.89	-	-	-	-
Spermine	11.77	14.52	16.27	14.58	N.D.	17.35	14.19	2.27	-	-	-	-
Spermidine	55.79	45.38	47.63	8.22	N.D.	N.D.	49.60	5.48	-	-	-	-

N.D.: not detected.

3. The term “hypusination” has to be introduced in more detail. Does there already exist any published data, which might suggest a potential role of eIF5A in IBD pathogenesis? This aspect should be included in the discussion.

Following this suggestion, we mentioned “hypusination” in the revised manuscript (lines 33–35) as follows: “Spermidine serves as a substrate for hypusine, in which the aminobutyl group of spermidine is attached to a specific lysine residue. Hypusination of eukaryotic translation initiation factor 5A (eIF5A) is required for its activation”

To our knowledge, the involvement of hyp-eIF5A in the IBD pathogenesis has not been directly explored. In Crohn’s disease patients, the expression of ornithine decarboxylase (ODC), a rate-limiting enzyme in polyamine biosynthesis, is downregulated in the inflamed region of the colon compared with that in the uninflamed region (Ricci et al. *Eur J Gastroenterol Hepatol*, 903-904. 1999, Krzystek-Korpacka et al. *Int J Mol Sci*. 21:1641. 2020). We discussed this issue in the revised manuscript (lines 312–316) as follows: Recent studies exhibited that ODC expression was downregulated in the inflamed region compared with that in the uninflamed region of the colonic mucosa in patients with Crohn’s disease. Thus, decreased endogenous polyamine synthesis might be involved in the development and/or exacerbation of inflammatory bowel disease (IBD).

4. The representative Western blot image depicted in Figure 2f shows only a minor regulation of hyp-eIF5A in MCHA-exposed organoids, which is hard to recognize at all. Are the authors

able to prove that this minor regulatory effect indeed functionally underlies the impaired growth of organoids?

MCHA dose-dependently decreased the hypusinated eIF5A, with an approximately 40% reduction at a higher dose (Fig. 2f). Additionally, 15 μ M GC7, a hypusination inhibitor, reduced the hypusinated eIF5A by approximately 50% (Fig. 2i), which sufficiently suppressed organoid growth (Fig. 2h). Furthermore, we treated the *E. coli*-WT-associated gnotobiotic mice with GC7, which reduced hypusinated eIF5A levels in the CECs by only 25% (Fig. 2j) but significantly suppressed CEC proliferation. Therefore, even a 40% MCHA-induced reduction in hypusinated eIF5A may affect organoid growth. Fig. B shows the western blot of the spermidine- and MCHA-treated organoids. We included these data in the revised manuscript.

Fig. B Western blot analysis of hyp-eIF5A in colonic organoids treated with MCHA and spermidine.

Updated Fig. 2j, k:

5. There is a discrepancy in Figure 2g between the depicted representative Western blot image and the summarizing graph: The image seems to indicate higher levels of hyp-eIF5A in epithelial cells of SK930-strain-associated mice than in epithelial cells of GF mice, which is not reflected by the summarizing graph.

We understand the concern that the difference in hyp-eIF5A was seemingly marginal. Therefore, we conducted the same experiments again and confirmed that hyp-eIF5A was attenuated in the CECs of the SK930-associated mice compared with those of the WT-associated mice (Fig C). We included part of Fig. C (in the rectangle) as a representative western blot image in the revised manuscript.

Fig. C. Re-examination of Western blot analysis of hyp-eIF5A in the CECs of the gnotobiotic mice.

Updated Fig 2g:

6. In Figure 2h, the authors demonstrate that the presence of GC7 (inhibitor of hyp-eIF5A synthesis) resulted in a decreased growth of organoids. It would be important to confirm the successful inhibition of hyp-eIF5A in this experimental setting. Are the authors able to exclude that the observed effect on organoid growth might be due to any unspecific toxic effect of the inhibitor?

In the experiment shown in Fig. 2h in the original manuscript, we added GC7 from the first day of colonic organoid culturing and observed an inhibition of organoid growth. However, we could not obtain proteins from the GC7-treated organoids because of insufficient growth. Therefore, we conducted an additional experiment to confirm the inhibitory effect of GC7 on hyp-eIF5A in the organoids. After culturing the organoids for 3 days without GC7, the organoids were treated with GC7 for 2 days. A higher dose (15 μ M) of GC7 significantly decreased the hyp-eIF5A levels in association with growth inhibition, suggesting that hyp-eIF5A deficiency arrested the organoid growth. We included these data in Fig. 2h, i:

Whether the growth arrest by GC7 resulted from hyp-eIF5A deprivation or an unspecific cytotoxic effect of GC7 was difficult to distinguish in this experimental setting. However, Lou et al. reported that GC7 induces little cytotoxicity up to 20 μ M, although high concentrations (50–100 μ M) of GC7 significantly inhibit the viability of hepatocellular carcinoma cells (Lou et al. *Exp Cell Res.* 319, 2708-2717, 2013). Therefore, we believe that the unspecific cytotoxic effects were minimal under our culture conditions (5–15 μ M). This information is included in the revised manuscript (lines 193–197): “Lou et al. reported that GC7 exerts little cytotoxicity up to 20 μ M, whereas higher GC7 concentrations (50–100 μ M) significantly inhibit hepatocellular carcinoma cell viability. Therefore, we believe that the growth arrest of colonic organoids by 15 μ M of GC7 did not result from the cytotoxicity of GC7.”

7. The authors hypothesize that it might be possible that exogenous polyamine may also regulate CX3CR1^{high} Ly6C⁻ macrophage differentiation through the metabolic reprogramming to OXPHOS by hypusinated eIF5A. This is very speculative. In order to meet the requirements of a high-ranking publication, this hypothesis needs to be validated by experimental analyses. Are the authors able to show altered hyp-eIF5A levels in BMDMs after exposure to putrescine?

Are they able to detect an altered expression of OXPPOS-related proteins in BMDMs cultured in the presence/absence of putrescine? Are these potential effects dependent on the presence of hypusinated eIF5A? Are BMDMs able to uptake exogenous putrescine?

Thank you for raising this critical issue. To determine whether BMDMs uptake exogenous polyamines, we treated BMDMs with 100 μ M 15 N₂-labelled putrescine for 24 hours and analysed the frequency of the isotope-labelled putrescine and its derivative, spermidine, in the intracellular polyamine pool. Isotope-labelled putrescine and spermidine occupied approximately 81.5% and 60.6% , respectively, of the corresponding polyamines (Fig. 4j, k in the revised manuscript). Thus, BMDMs actively used exogenous putrescine to produce spermidine (line 254–260). These data are included in Fig. 4j, k:

Following the suggestion, we assessed the effect of exogenous putrescine on hyp-eIF5A and OXPPOS. We treated bone marrow cells with macrophage colony-stimulating factor in the presence or absence of DFMO and putrescine. Hypusination of eIF5A was upregulated in BMDMs stimulated with IL-4, but not with LPS + IFN- γ , consistent with previous studies (Puleston et al. *Cell Metab.* 30, 352-363, 2019). While DFMO treatment reduced hyp-eIF5A, exogenous putrescine rescued this reduction (Supplementary Fig. 6)

We further analysed whether polyamines cause metabolic reprogramming in BMDMs. The expressions of OXPPOS complex proteins CI–CV were assessed via western blotting. CI, CII and CIV expressions decreased in the DFMO-treated macrophages. These alterations were similar to those in BMDMs stimulated with IL-4 (Puleston et al. *Cell Metab.* 30, 352-363, 2019). Exogenous putrescine restored the CI, CII and CIV expressions in the DFMO-treated cells. We included these data in Supplementary Fig. 6:

We discussed this issue in the revised manuscript (lines 263–280) as follows: “To assess the effect of exogenous putrescine on hyp-eIF5A and OXPPOS, we treated bone marrow cells with macrophage colony-stimulating factor and the presence or absence of DFMO and putrescine. Hypusination of eIF5A was upregulated in macrophages stimulated with IL-4, but not in those stimulated with LPS + IFN- γ , which is similar to the results of previous studies (Supplementary

Fig. 6). DFMO treatment decreased hyp-eIF5A in BMDMs, and exogenous putrescine rescued this DFMO-induced reduction in hyp-eIF5A. Furthermore, exogenous putrescine treatment without DFMO slightly increased the hyp-eIF5A (Supplementary Fig. 6). We further analysed whether polyamines cause metabolic reprogramming in BMDMs. The expressions of OXPHOS complex proteins CI–CV were assessed via western blotting. CI, CII and CIV were upregulated in macrophages stimulated with IL-4, but not in those stimulated with LPS + IFN- γ , consistent with previous studies (Supplementary Fig. 6). Furthermore, CI, CII and CIV expressions were decreased in DFMO-treated macrophages, and exogenous putrescine restored this suppression. OXPHOS complex protein expression in macrophages treated only with putrescine resembled that in macrophages stimulated with IL-4 (Supplementary Fig. 6). Thus, exogenous putrescine induced metabolic reprogramming similar to that of IL-4 stimulation.”

8. The described observation that WT-strain-associated mice were more resistant to DSS-induced colitis than SK930-strain-associated mice is somehow in accordance with the following recently published study, which described that spermidine protects mice during DSS-induced colitis: Gobert AP, Al-Greene NT, Singh K, et al. Distinct Immunomodulatory Effects of Spermine Oxidase in Colitis Induced by Epithelial Injury or Infection. Front Immunol. 2018;9:1242. Thus, it is very important to mention and discuss this study. Moreover, another recent study indicated that polyamines promote the differentiation of regulatory T cells (Carriche GM, Almeida L, Stüve P, et al. Regulating T cell differentiation through the polyamine spermidine. J Allergy Clin Immunol. 2020). Are the authors able to confirm experimentally that the identified effects of polyamines on intestinal epithelial cells and/or macrophages indeed underlie the observed differences in the severity of colitis between WT-strain-associated mice and SK930-strain-associated mice?

Thank you for providing this important information. We cited the study of Gobert et al. and discussed the protective effect of spermidine on DSS-induced colitis (line 299–303). We added the following sentence: “Interestingly, a recent study demonstrated that spermine oxidase deficiency reduced colonic spermidine levels, which exacerbated lethality and mucosal inflammation, in a DSS-induced colitis model. This observation supports the importance of polyamine metabolism in maintaining gut immune homeostasis.”

Carriche et al. found that spermidine promotes differentiation into CD4⁺Foxp3⁺ T cells *in vitro*. Therefore, we analysed Foxp3⁺ regulatory T (Treg) cells in the colonic lamina propria of GF and gnotobiotic mice associated with either the WT or SK930 strains. However, the Treg cell frequencies did not significantly differ among the three groups. These data are included in Updated Supplementary Fig. 5c:

CX₃CR1^{low}Ly6C⁺ and NOS2⁺ M1-like subsets were less abundant in the cLPs of the WT strain-associated mice than in the SK930-strain-associated mice after DSS treatment (Fig. 5g–j and Supplementary Fig. 7d–f in the revised manuscript). Because DSS colitis is mainly mediated

by these inflammatory myeloid cell subsets, we think that colitis suppression in the WT-strain-associated mice is attributed to regulation of the M1/M2 balance rather than to the induction of Treg cells. We discussed this issue in the lines 303–309 in the revised manuscript.

Minor concerns:

- *The assignment of b and c in the legend of Supplementary Figure 3 is not correct.*

We corrected the assignments of b and c in the legend.

- *In Figure 3d, an isotype control staining should be provided in order to confirm the correct gating of the NOS2-expressing cell population.*

To confirm the gating of NOS2- and Arg1-positive cells, we used isotype control antibodies as negative controls in the flow cytometry and reanalysed the flow cytometry data (Fig. 4d). We confirmed that both the NOS2⁺ and Arg1⁺ cell gates are correct (Fig. D).

Fig. D Dot plot image of flow cytometry staining with isotype control (left panel), and staining with Arg1-FITC and NOS2-PE (right panel).

Updated Fig. 4d–g:

REVIEWERS' COMMENTS

Reviewer #1 (Remarks to the Author):

The authors addressed all my comments; in conjunction with the suggestions made by the other reviewers, the paper has greatly improved.

Reviewer #2 (Remarks to the Author):

I think the authors have done a very thorough job of answering my questions and I am satisfied with their responses.

Reviewer #3 (Remarks to the Author):

The authors have carefully addressed most of my concerns. It is a solid and interesting paper now. Some prior art on the subject has been overlooked and should be cited before publication:

Spermidine improves gut barrier integrity and gut microbiota function in diet-induced obese mice
Lingyan Ma , Yinhua Ni , Zhe Wang , Wenqing Tu , Liyang Ni , Fen Zhuge , Aqian Zheng , Luting Hu , Yufeng Zhao , Liujie Zheng & Zhengwei Fu
Gut Microbes, 2020

Induction of autophagy by spermidine promotes longevity.

Eisenberg T, Knauer H, Schauer A, Büttner S, Ruckstuhl C, Carmona-Gutierrez D, Ring J, Schroeder S, Magnes C, Antonacci L, Fussi H, Deszcz L, Hartl R, Schraml E, Criollo A, Megalou E, Weiskopf D, Laun P, Heeren G, Breitenbach M, Grubeck-Loebenstien B, Herker E, Fahrenkrog B, Fröhlich KU, Sinner F, Tavernarakis N, Minois N, Kroemer G, Madeo F. Nat Cell Biol. 2009 Nov;11(11):1305-14. doi: 10.1038/ncb1975. Epub 2009 Oct 4.

Autophagy is a critical regulator of memory CD8(+) T cell formation.

Puleston DJ, Zhang H, Powell TJ, Lipina E, Sims S, Panse I, Watson AS, Cerundolo V, Townsend AR, Klenerman P, Simon AK. Elife. 2014 Nov 11;3:e03706. doi: 10.7554/eLife.03706. PMID: 25385531 Free PMC article.

Hypusination Orchestrates the Antimicrobial Response of Macrophages.

Gobert AP, Finley JL, Latour YL, Asim M, Smith TM, Verriere TG, Barry DP, Allaman MM, Delgado AG, Rose KL, Calcutt MW, Schey KL, Sierra JC, Piazuelo MB, Mirmira RG, Wilson KT. Cell Rep. 2020 Dec 15;33(11):108510. doi: 10.1016/j.celrep.2020.108510.

Reviewer #4 (Remarks to the Author):

In the revised version of the manuscript and in the provided point-by-point reply, the authors have convincingly and adequately addressed all the issues raised. There are no further concerns from my side.

NCOMMS-20-10409-A

"Symbiotic polyamine metabolism regulates epithelial proliferation and macrophage differentiation in the colon."

Reviewer #1 (Remarks to the Author):

The authors addressed all my comments; in conjunction with the suggestions made by the other reviewers, the paper has greatly improved.

Response:

We thank the reviewer #1 for finding great improvement in our revised manuscript.

Reviewer #2 (Remarks to the Author):

I think the authors have done a very thorough job of answering my questions and I am satisfied with their responses.

Response:

We thank the reviewer #2 for evaluating our revised manuscript positively.

Reviewer #3 (Remarks to the Author):

The authors have carefully addressed most of my concerns. It is a solid and interesting paper now. Some prior art on the subject has been overlooked and should be cited before publication:

Response:

We thank the reviewer #3 for positive response and important suggestion. We have cited the four papers in the revised manuscript.

Reviewer #4 (Remarks to the Author):

In the revised version of the manuscript and in the provided point-by-point reply, the authors have convincingly and adequately addressed all the issues raised. There are no further concerns from my side.

Response:

We thank the reviewer #4 for evaluating our revised manuscript positively.